# Genomic evidence for rediploidization and adaptive evolution following the whole-genome triplication

Xiao Feng [1,4], Qipian Chen [1,2,4], Weihong Wu[1], Jiexin Wang[1], Guohong Li[1], Shaohua Xu[1], Shao Shao[1], Min Liu[1], Cairong Zhong[3], Chung-I Wu[1], Suhua Shi [1] ✉ & Ziwen He [1] ✉

Whole-genome duplication (WGD), or polyploidy, events are widespread and significant in the evolutionary history of angiosperms. However, empirical evidence for rediploidization, the major process where polyploids give rise to diploid descendants, is still lacking at the genomic level. Here we present chromosome-scale genomes of the mangrove tree *Sonneratia alba* and the related inland plant *Lagerstroemia speciosa*. Their common ancestor has experienced a whole-genome triplication (WGT) approximately 64 million years ago coinciding with a period of dramatic global climate change. *Sonneratia*, adapting mangrove habitats, experienced extensive chromosome rearrangements post-WGT. We observe the WGT retentions display sequence and expression divergence, suggesting potential neo- and sub-functionalization. Strong selection acting on three-copy retentions indicates adaptive value in response to new environments. To elucidate the role of ploidy changes in genome evolution, we improve a model of the polyploidization–rediploidization process based on genomic evidence, contributing to the understanding of adaptive evolution during climate change.

The origin and radiation of flowering plants (angiosperms) in the mid-Cretaceous was famously referred to by Charles Darwin as "an abominable mystery"[1,2]. Presently, angiosperms encompass over 90% of all living plant species, with approximately 350,000 known species, making them the most successful land plants on Earth (www.theplantlist.org). There is growing consensus that whole-genome duplication (WGD) events, also known as polyploidy, have played a widespread and significant role in the evolutionary history of angiosperms[3–9]. Early WGD events in plants can be traced back to the common ancestors of extant seed plants and angiosperms, respectively[10]. Furthermore, core eudicots, a major clade within angiosperms, experienced a well-known paleo-hexaploidization

event[11]. WGDs have also occurred in various lineages, even recurrently, like *Arabidopsis*[12,13], soybean[14], carrot[15], and *Utricularia*[16]. One specific type of WGD, known as whole-genome triplication (WGT) or hexaploidy, originated through hybridization between tetraploid and diploid species[17–19]. Genomic data have revealed at least 18 independent WGT events in eudicots[11,17,20–34], indicating a prevalence higher than previously assumed (Supplementary Fig. 1). Despite the challenges that have emerged since the WGD, such as stable chromosome segregation, detrimental ecological interactions with diploid progenitors, and minority cytotype exclusion[35,36], the polyploidy events observed in plants highlight their evolutionary potential. Experimental and simulation studies have supported the adaptive potential of

[1]State Key Laboratory of Biocontrol and Guangdong Provincial Key Laboratory of Plant Resources, School of Life Sciences, Southern Marine Science and Engineering Guangdong Laboratory (Zhuhai), Sun Yat-sen University, 510275 Guangzhou, China. [2]Shenzhen Branch, Guangdong Laboratory of Lingnan Modern Agriculture, Genome Analysis Laboratory of the Ministry of Agriculture and Rural Affairs, Agricultural Genomics Institute at Shenzhen, Chinese Academy of Agricultural Sciences, 518120 Shenzhen, China. [3]Hainan Academy of Forestry (Hainan Academy of Mangrove), 571100 Haikou, China. [4]These authors contributed equally: Xiao Feng, Qipian Chen. ✉e-mail: lssssh@mail.sysu.edu.cn; heziwen@mail.sysu.edu.cn

polyploidy, especially in the face of dramatic and fluctuating environmental conditions[37–39]. Overall, polyploidy has been recognized as a major driving force behind evolutionary adaptation and diversification[4,5].

Plants have experienced periods of global climate change, and genomic resources offer an opportunity to better understand the dynamics of plant evolution during such global climate changes[40–42]. The relationship between WGD and the success of plant lineages is an intriguing topic[43–46]. Previous studies have revealed several instances of WGD occurring independently during three periods of climatic instability and environmental perturbations: the Early Cretaceous around 120 million years ago (Mya)[47], the K-Pg boundary around 65 Mya[48], and the Miocene–Pliocene (<20 Mya)[9,49]. These WGD events may have provided a buffer for plants and facilitated their survival and adaptation to rapidly changing environments by increasing genomic plasticity and genotypic combinations.

Mangroves have successfully adapted to extreme intertidal zones, bridging terrestrial and marine ecosystems, evolving a series of adaptive traits, such as salt tolerance, aerial roots, and viviparous embryos[50–52]. They are attractive ecological model systems to investigate adaptive evolution. Prior to colonizing their new habitat, several mangrove species independently experienced WGD events[53–57]. Nevertheless, almost all mangrove species are currently considered diploids (Supplementary Data 1), indicating the importance of the rediploidization process in ancient polyploids. Rediploidization involves redundancy reduction, coordination of subgenomic function, and chromosome fractionations, ultimately leading to the establishment of modern diploid descendants cytogenetically and potentially contributing to plant adaptation[58,59]. Despite its significance, the rediploidization process in ancient polyploid plants remains poorly understood. With advancements in genome sequencing and assembly technologies, high-quality chromosome-scale genomes provided an opportunity to reconstruct ancestral genomes and infer the trajectory of plant genome evolution[60,61]. We can now explore the process of rediploidization following polyploidization on a genomic scale.

In this work, we present two chromosome-scale genomes of Lythraceae plants: the mangrove tree *Sonneratia alba* (Supplementary Fig. 2) and related inland plant *Lagerstroemia speciosa* (Supplementary Fig. 3), as a part of the worldwide mangrove genomes project[62]. Through comprehensive analyses, we trace the evolutionary history of genomes and investigate the polyploidization–rediploidization process and its implications for adaptive evolution in the face of global climate change.

## Results and discussion
### Genome sequencing, assembly, and annotation
We first utilized high-throughput chromosome conformation capture (Hi-C) technology to improve the genome of *S. alba*. This improvement builds upon our prior study utilizing PacBio Single-Molecule Real-Time (SMRT) sequencing and Illumina short reads sequencing[54], resulting in a chromosome-scale assembly (Supplementary Table 1). The assembled genome derived from anchored contigs was 204.46 Mb, aligning closely with the genome size estimated through k-mer-based analysis (211.67 Mb). It comprised 12 chromosomes (97.60% of all sequences) and 40 unanchored scaffolds. The N50 value notably increased from 5.52 Mb to 15.69 Mb (Table 1). Additionally, we de novo assembled the genome of the closely related inland woody plant *L. speciosa* by incorporating high-depth PacBio SMRT sequencing, Illumina short reads sequencing, and Hi-C technologies (Supplementary Table 1). The assembled genome of *L. speciosa* was 319.66 Mb, with an N50 value reaching 12.74 Mb, consistent with the estimated genome size (361 Mb by flow cytometry and 340.46 Mb by k-mer-based analysis). It comprised 24 chromosomes, encompassing 98.08% of all sequences (Table 1). The assembled genomes of *S. alba* and *L. speciosa* both showed high congruence because of their strongest interaction

**Table 1 | Statistics for *Sonneratia alba* and *Lagerstroemia speciosa* genomes**

| Genome features | Sonneratia alba | Lagerstroemia speciosa |
|---|---|---|
| Sequencing methods | Illumina + PacBio + HiC | Illumina + PacBio + HiC |
| Sequencing reads | 32.99 Gb + 28.36 Gb + 103.88 Gb | 41.16 Gb + 95.60 Gb + 54.19 Gb |
| Assembled genome size | 204.46 Mb | 319.66 Mb |
| Anchored size | 199.55 Mb (97.60%) | 313.51 Mb (98.08%) |
| Anchored gene number | 25,126 | 30,323 |
| GC content | 41.77% | 40.40% |
| Number of chromosomes | 2n = 24 | 2n = 48 |
| Number of scaffolds | 52 | 629 |
| N50 length | 15.69 Mb | 12.74 Mb |
| N90 length | 12.96 Mb | 10.54 Mb |
| Longest sequence length | 22.93 Mb | 17.34 Mb |
| Gap content | 0.05% | 0.02% |

signals from the Hi-C data clustered at the expected diagonal region (Fig. 1a and Supplementary Fig. 4). The gene prediction process involved a comprehensive approach, combining ab initio, homology-based and RNA-seq-assisted strategies. The integration of these predictions through EvidenceModeler resulted in the identification of non-redundant and consensus gene models for the *S. alba* and *L. speciosa* genomes (see Methods for details). This unveiled a total of 25,284 (Supplementary Fig. 5) and 30,497 (Supplementary Fig. 6) protein-coding genes, respectively, characterized by high completeness (Supplementary Table 2). Moreover, 99.38% and 99.43% of them were categorized into chromosomes, respectively. The presence of syntenic blocks between the two genomes further supported their quality as chromosome-scale assemblies (Fig. 1b). These high-quality genomes can supplant earlier assemblies, serving as valuable references for genomic and evolutionary studies in plants (Supplementary Note 1).

### Less TE accumulation in the mangrove
The mangrove species have small genome sizes compared with inland relatives[63]. Repetitive sequences are the primary determinant of plant genome size, and transposable elements (TEs) are the predominant components of repetitive elements[64,65]. First, we observed that *S. alba* has fewer chromosomes compared to *L. speciosa* (Fig. 1b). We estimated that 20.95% (43 Mb) of the *S. alba* genome consists of TE sequences, while 36.50% (117 Mb) in the *L. speciosa* genome and higher TE contents in other relatives (Supplementary Fig. 7 and Supplementary Table 3). The long terminal repeat retrotransposons (LTR-RTs), typical class I TEs, usually have much copy number and large size in plant genomes, contributing significantly to genome size growth[66]. The intact LTR-RTs were further classified as *Copia* and *Gypsy* element families, and their insertion time distributions were examined. We found that *S. alba* has much lower recent LTR-RT insertion rates than relatives in Myrtales, especially in the *Copia* element family (Supplementary Fig. 8). Overall, the mangrove species *S. alba* maintains a smaller genome size, fewer chromosomes, lower accumulation of TEs, and a reduced rate of LTR-RT insertion, resulting in a more simplified genome.

### WGT coinciding with dramatic global climate change
With the availability of chromosome-scale reference genomes in Lythraceae, we revisited the origin of *Sonneratia*, the significant taxon within the mangrove ecosystem. We reconstructed the phylogeny

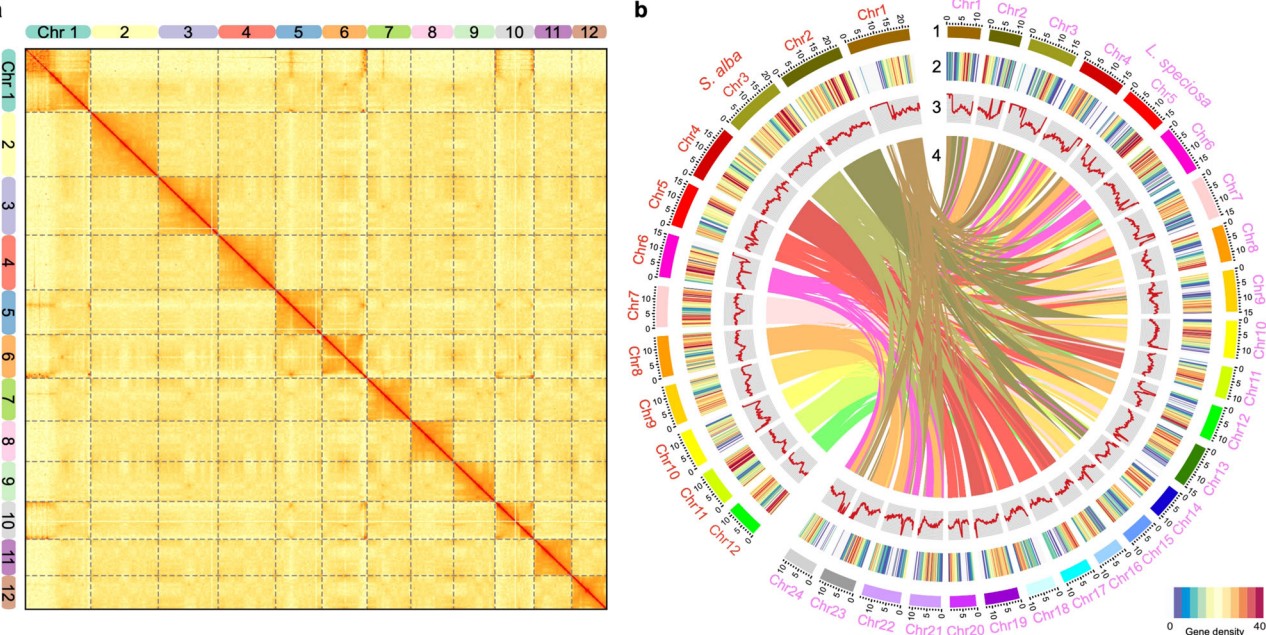

**Fig. 1 | Genomic architecture of *Sonneratia alba* and relatives. a** Hi-C interactive heatmap of the genome-wide organization of *S. alba*. The deeper red means the stronger interaction between the DNA regions. Chr: chromosome. **b** Circos plot of the *S. alba* and *L. speciosa* genomes. Concentric circles, from outer to inner, show (1) pseudo-chromosome (Mb), (2) gene density, (3) GC content (34.01–57.76% per 200 Kb), and (4) syntenic block. Source data are provided as a Source Data file.

among three Lythraceae species (*S. alba*, *L. speciosa*, and *Punica granatum*) and four other species (*Eucalyptus grandis*, *Arabidopsis thaliana*, *Vitis vinifera*, and *Nelumbo nucifera*) with available pseudo-chromosome scale genome data. The tree topology was inferred using RAxML-NG with the GTR + GAMMA + I model based on 1,963 ortholo-gous single-copy gene groups (Supplementary Fig. 9), and the diver-gence time was estimated using MCMCTREE from the PAML package with two reliable calibrations (see Methods for details). The divergence times were consistent with a previous study (Supplementary Table 4)[67]. Additionally, our estimation suggests that the mangrove *S. alba* diverged from the closely related inland woody plant *L. speciosa* around 57.79 Mya, while the common ancestor of them diverged from the same family plant *P. granatum* around 67.82 Mya (Fig. 2a). We further constructed a larger-scale phylogenetic tree, incorporating 42 sequenced angiosperms along with the gymnosperm *Gnetum mon-tanum* (as an outgroup), to reflect the positions of these plants within Lythraceae (Supplementary Fig. 10).

Whole-genome duplication (WGD), or polyploidy, events have played a significant role in the evolutionary history of angiosperms, aiding in their survival during periods of dramatic environmental changes[4,9,43]. WGD events can provide a substantial amount of genetic material for adaptation. In this study, we utilized a combination of synteny, Ks-base, and phylogenetic approaches (Supplementary Fig. 11) to confirm that *S. alba* and *L. speciosa* underwent a whole-genome triplication (WGT) event prior to their divergence from a common ancestor (Fig. 2a). Initially, we scanned the genomes of three Lythraceae plants, namely *S. alba*, *L. speciosa*, and *P. granatum*, using BLASTP and MCScanX. We identified 164 syntenic block pairs com-prising 5,999 gene pairs in *S. alba*; 486 syntenic block pairs comprising 12,180 gene pairs in *L. speciosa*; and 219 syntenic block pairs com-prising 3,333 gene pairs in *P. granatum*. The presence of extensive syntenic block pairs indicated past polyploidy events. Subsequently, we calculated synonymous substitution rates (Ks) between paralogous genes in each genome. The Ks distribution revealed recent peaks in *S. alba* and *L. speciosa*, but not in *P. granatum* (Fig. 2b), suggesting that *P. granatum* did not experience the polyploidy events. Within the Ks peaks range, we identified 584 three-copy retention groups in *S. alba*

and 943 in *L. speciosa* (Supplementary Figs. 12–14), indicating that the polyploidy event in these species was a hexaploidy (whole-genome triplication, WGT) event. This finding was further supported by genome-wide syntenic regions between *S. alba* and *P. granatum*, as well as *L. speciosa*, and *P. granatum* (Fig. 2c and Supplementary Fig. 15). Furthermore, we presented an expected signature of the whole-genome triplication event through collinear genes in the modern genome (Supplementary Fig. 16). While the Ks peak appears slightly different between *S. alba* and *L. speciosa*, we performed gene tree reconstructions of the syntenic gene groups and confirmed that the WGT event occurred prior to the divergence between *Sonneratia* and *Lagerstroemia* (Fig. 2d). The placement of the WGT event was also validated using the multi-taxon paleopolyploidy search (MAPS) ana-lysis and corresponding simulations (Supplementary Fig. 17). This multipronged approach allows us to overcome the challenges posed by divergent evolutionary rates in different plants, enabling the iden-tification of more accurate features and positions of polyploidy events[23,68–70].

Extrapolating from the divergence time in Lythraceae, we further estimated that the shared WGT event of *Sonneratia* and *Lagerstroemia* occurred around 64 Mya (Fig. 2a, see Methods for details), slightly after the divergence from *P. granatum*. This WGT event coincided with a brief period of dramatic global climate change resulting from a large asteroid collision with the Earth, known as the Cretaceous-Paleogene (K-Pg) boundary, which took place around 66 Mya[48]. Polyploidy events play a significant role in reshaping gene regulatory networks in response to environmental stresses[9,71]. A series of ancient WGD events occurred independently in numerous plant lineages around the K-Pg boundary[43,45,49]. These events served as a buffer for plants, enhancing their ability to survive and adapt to rapidly changing environments by increasing genomic plasticity and generating diverse genotypic com-binations. We suggest that the WGT events may have contributed to the survival of plants during the extinction event. Not only that, at approximately 55 Mya, there was a significant global temperature increase (warming by ~6 °C within ~20,000 years) and a rise in eustatic sea levels, known as the Paleocene-Eocene Thermal Maximum (PETM)[72]. The combination of sea level rise, mass extinction, and the

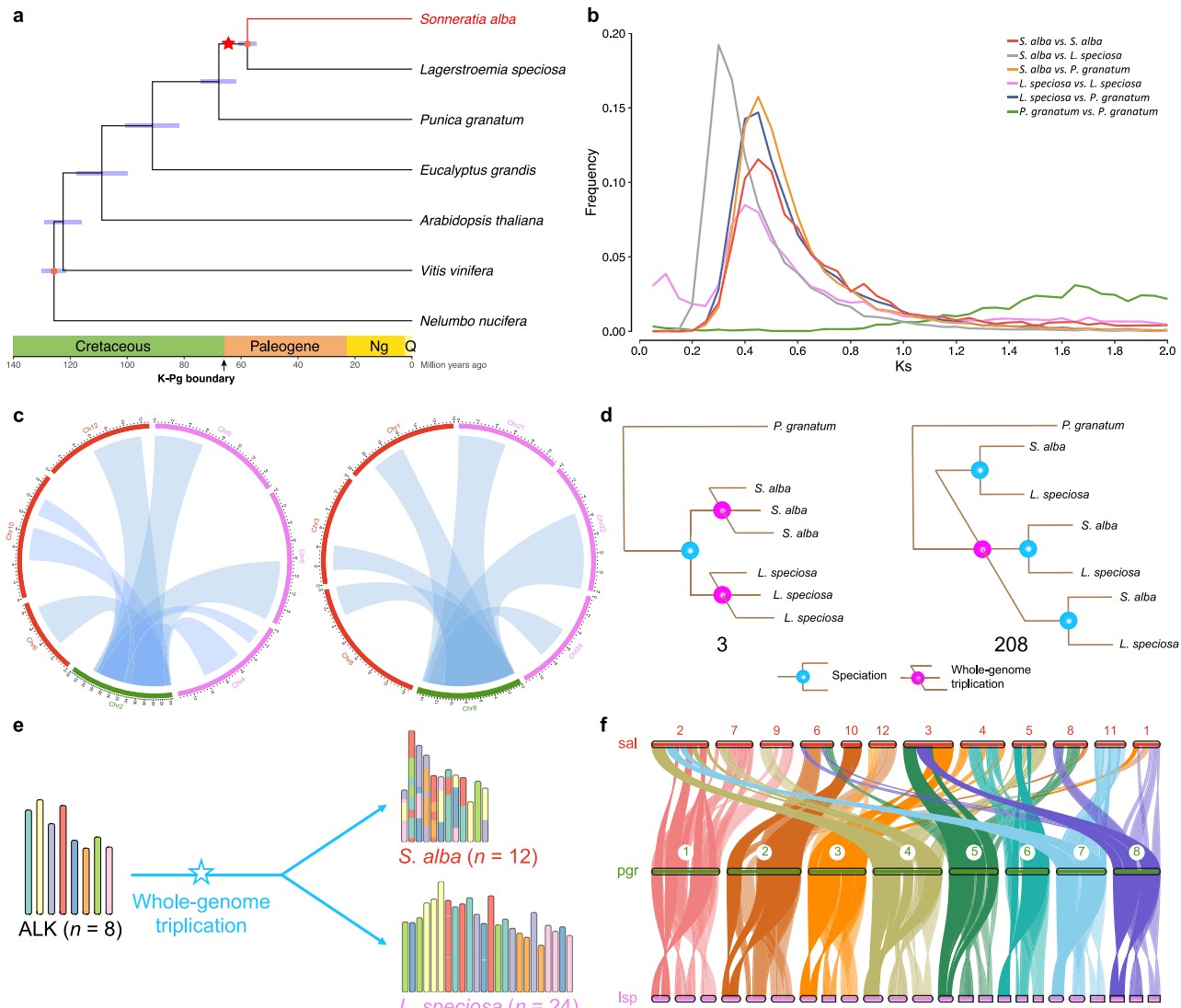

**Fig. 2 | The whole-genome triplication (WGT) event is shared in *S. alba* and *L. speciosa*. a** Phylogenetic tree of seven eudicots, including *S. alba* and relatives. Blue node bars are 95% confidence intervals. Red nodes indicate two fossil calibration nodes. The star represents the WGT event. The occurrences of the K-Pg boundary and PETM are indicated by the arrows on the timeline. **b** Ks distribution between paralogous genes within the same species and orthologous genes from pairs of species. **c** Synteny between the homologous regions of *S. alba* (red), *L. speciosa* (pink), and *P. granatum* (green). It reflects the overall synteny relationship with a 3:1 ratio between *S. alba vs. P. granatum*, *L. speciosa vs. P. granatum*, respectively. The representation showcases partial regions of the genomes. **d** Numbers of homologous gene groups supporting different scenarios on the order of speciation and WGT events in *Sonneratia* and *Lagerstroemia*. **e** Chromosome evolution following WGT from the ALK (the ancestral Lythraceae karyotype). The star represents the WGT event. **f** Macrosynteny patterns among the three Lythraceae plants. "sal" represents *S. alba*, "pgr" represents *P. granatum*, and "lsp" represents *L. speciosa*.

WGT event potentially provided an opportunity in environmental and genetic aspects for offshore woody plants to develop a series of highly specialized traits (such as salt tolerance and aerial roots) to survive, leading to the emergence of the mangrove *Sonneratia*.

Genome evolution is a long-term and dynamic process. Early WGD events (ζ, ε, γ) occurred hundreds of millions of years ago[10,11] and their corresponding collinearity has faded with time or been influenced by subsequent WGD events. Plants that have undergone recent WGD events within the past 20 million years still possess numerous redundant regions in the genomes. Therefore, the WGT event in *Sonneratia* (~64 Mya) provides a valuable opportunity to study the polyploidization–rediploidization process in angiosperms (Fig. 2a). Furthermore, by integrating appropriate genomic data, we positioned the WGT event within a narrow time window between two close speciation events, whose pattern is similar to the γ-WGT event associated with the early diversification of core eudicots.

## Chromosome evolution following the WGT event

Many ancient polyploidy events have been followed by striking reductions in duplicated redundancy and chromosome number[59]. For example, *Utricularia gibba*, despite having a small plant genome, has a haploid chromosome number ($n$) of 14, yet it has undergone three WGD events since the well-known γ event shared by core eudicots[16]. If we exclusively consider polyploidy, the haploid chromosome number of *Utricularia gibba* would be $7 \times 3 \times 2 \times 2 \times 2$ or $n = 168$, based on the ancestral chromosome number ($n = 7$) before experiencing γ-WGT event[73]. Conducting a chromosome-scale comparative investigation among the Lythraceae plants allowed us to explore the paleo-history following the WGT event. Our analysis inferred that the chromosome number of the common ancestor of *Sonneratia* and *Lagerstroemia* is $n = 24$ (post-WGT) and $n = 8$ (pre-WGT) (Supplementary Fig. 18). Additionally, the chromosome number of the common ancestor of the three Lythraceae plants is $n = 8$, which is the same as the chromosome number in *P. granatum*.

To gain further insights into the evolutionary history of chromosomes, we reconstructed the ancestral Lythraceae karyotype (ALK) using WGDI based on adjacent conserved collinear blocks. Our evolutionary scenario suggests that the ALK of *S. alba*, *L. speciosa*, and *P. granatum* genomes consisted of eight proto-chromosomes with 18,885 proto-genes. As shown in Fig. 2e, the ancestor underwent a WGT event and subsequently experienced chromosomal rearrangements to attain their modern genome structure. The chromosome origin of *S. alba* appears more intricate than that of *L. speciosa*. *S. alba*'s chromosomes underwent a greater number of fission and fusion events compared to *L. speciosa*, although intra-chromosomal inversions were common in the chromosome histories of both species (Fig. 2e and Supplementary Fig. 19). Due to the complexity of chromosome evolutionary history in *S. alba*, we illustrated it using reciprocally translocated chromosome arms (RTA), end-to-end joining (EEJ), nested chromosome fusion (NCF) events, fission events, and chromosome inversions to depict a probable karyotype evolution (Supplementary Fig. 20).

Although the reconstructed ancestral karyotype is highly likely to possess a structure very similar to the true ancestral genome, it may not be entirely identical[60]. Furthermore, we performed synteny analysis among the modern genomes of the three Lythraceae species and confirmed numerous chromosome rearrangements (Fig. 2f). In contrast to intra-chromosomal inversions observed in related inland species, *S. alba* exhibited significant fission and fusion events (Supplementary Fig. 21). These findings indicate that the mangrove species has a reduced number of chromosomes and underwent more chromosomes rearrangements compared to its closely related inland species *L. speciosa*.

## Adaptation through polyploidization–rediploidization cycles

During periods of dramatic environment and climate change, newly formed polyploids can possess fitness advantages over diploids. This is supported by evidence that the persistence of WGD correlates with times of environmental and climate change, suggesting potential benefit for the WGD in the face of challenges[4,35,45,74–77]. Nevertheless, polyploids may also face substantial disadvantages, including redundant components, gene dosage imbalance, increased replication and metabolic costs, cellular mismanagement, and a higher propensity for polyploid mitosis and meiosis to produce aneuploid cells[35,58,77,78]. Despite these immediate challenges, some polyploid lineages have persisted and even thrived[79,80]. As climatic conditions stabilize and environmental conditions improve, polyploids may experience reduced fitness compared to diploids due to the accumulation of genetic load, increased mutational load, slower positive selection, and reduced growth rates[35,37,81,82]. Therefore, the process of rediploidization following polyploidization may be inevitable for polyploids, ultimately leading to modern descendants as normal diploids cytogenetically, generating important genetic and taxonomic diversity. In fact, nearly all angiosperms have undergone successive rounds of polyploidization and rediploidization process (Supplementary Fig. 22)[4,10,11,83]. Considering the potential role of ploidy changes in genome evolution, we improve a model based on genomic evidence and the previous studies[35,58,59,77,84,85]. This model explains the polyploidization–rediploidization process, elucidating the adaptive evolution during global upheavals and restoration (Fig. 3 and Supplementary Data 2). Specifically, rediploidization through redundancy reduction, gene divergence and chromosome rearrangement confers advantages, such as shortening DNA replication and the cell cycle, and reducing recombination of locally adapted alleles, thereby facilitating the survival of the mangrove in barren intertidal zones.

## Divergence of WGT retained genes in the mangrove genome

The differentiation of retained genes plays a crucial role in reducing gene redundancy and serves as a primary genetic basis for genome evolution. We observed that paralogous gene pairs generated by the WGT event in *S. alba* exhibited higher genetic divergences (Supplementary Fig. 23) and Ks values (Fig. 2b), indicating sequence differentiation. Besides the sequence divergence, expression divergence is also important. Therefore, we conducted transcriptome sequencing of leaf, root, flower, and fruit tissues of *S. alba* (Supplementary Fig. 24 and Supplementary Table 5) and employed the exact conditional test to investigate the expression divergence of WGT retained genes. We identified that approximately 58.04% to 64.57% of the paralogous gene pairs generated by the WGT were differentially expressed across these four tissues in the mangrove species (Supplementary Table 6). The different tissues harbored a similar number of differentially expressed gene pairs (DEGPs), with slightly higher numbers in leaf tissue and lower numbers in fruit tissue. These differentially expressed pairs belonged to 481–516 three-copy retention groups and 1937–2136 two-copy retention groups in different tissues (Supplementary Table 6). Moreover, we identified 1,789 gene pairs that showed differential expression across all four tissues (Supplementary Fig. 25). To investigate the functional roles of these DEGPs, we performed gene ontology (GO) enrichment analysis. The DEGPs in different tissues were predominantly enriched in the metabolic process and catalytic activity GO categories, while the DEGPs shared across all four tissues were enriched in more specific GO categories related to metabolic process, gene expression, biosynthetic process, mitochondrial envelope, and catalytic activity (Supplementary Data 3), which are critical for plant growth and adaptation. Similarly, we explored the expression divergence of WGT retained genes in the closely related inland plant *L. speciosa*. We also identified that around 60% of the paralogous gene pairs resulting from the WGT exhibited differential expression across four tissues in the related species (Supplementary Tables 7, 8), mirroring findings in the mangrove species. These results suggest the potential neo- and sub-functionalization of the retained genes following the polyploidization–rediploidization process.

## Strong selection in WGT retained gene groups

Polyploidy is widely recognized as a major source of novel genetic material, which can undergo mutation and selection to give rise to new or specialized functions to aid adaptation[86,87]. To assess the impact of both negative and positive selection on sites located in different-copy (one-, two-, three-copy) retention groups in the mangrove species, we used the DFE-alpha approach to estimate the distribution of fitness effects (DFE) of new mutations and the proportion of adaptive divergence ($\alpha$)[88–91], based on the folded site frequency spectrum (SFS) and divergence between *S. alba* and *S. apetala*. We also estimated constraint and selection effects using SnIPRE[92], as well as fixation index (FI)[93] for genes belong to different-copy retention groups to demonstrate the strength of selection (Fig. 4 and Supplementary Fig. 26). We observed that the one-copy retention group exhibited a lower proportion of strongly deleterious mutations ($N_e\mathcal{S} < -100$) and a higher proportion of slightly deleterious mutations ($-1 < N_e\mathcal{S} < 0$) compared to other copy retention groups (Fig. 4a and Supplementary Fig. 26a). The proportion of adaptive divergence ($\alpha$) at zero-fold nonsynonymous sites and neutral divergence ($\omega_a$) were both higher in the three-copy retention group (Fig. 4a and Supplementary Fig. 26a). They suggested that the strength of negative selection and positive selection were increased in turn from the one-copy to the three-copy retention groups, consistent with estimates of FI (Fig. 4b and Supplementary Fig. 26b). Furthermore, the three-copy retention group demonstrated a lower constraint effect and a higher selection effect compared to other copy retentions (Fig. 4c, d and Supplementary Fig. 26c, d). Collectively, these results indicate preferential retentions of three-copy genes following the polyploidization–rediploidization process, driven by strong selection and possessing potential adaptive value in response to new environments.

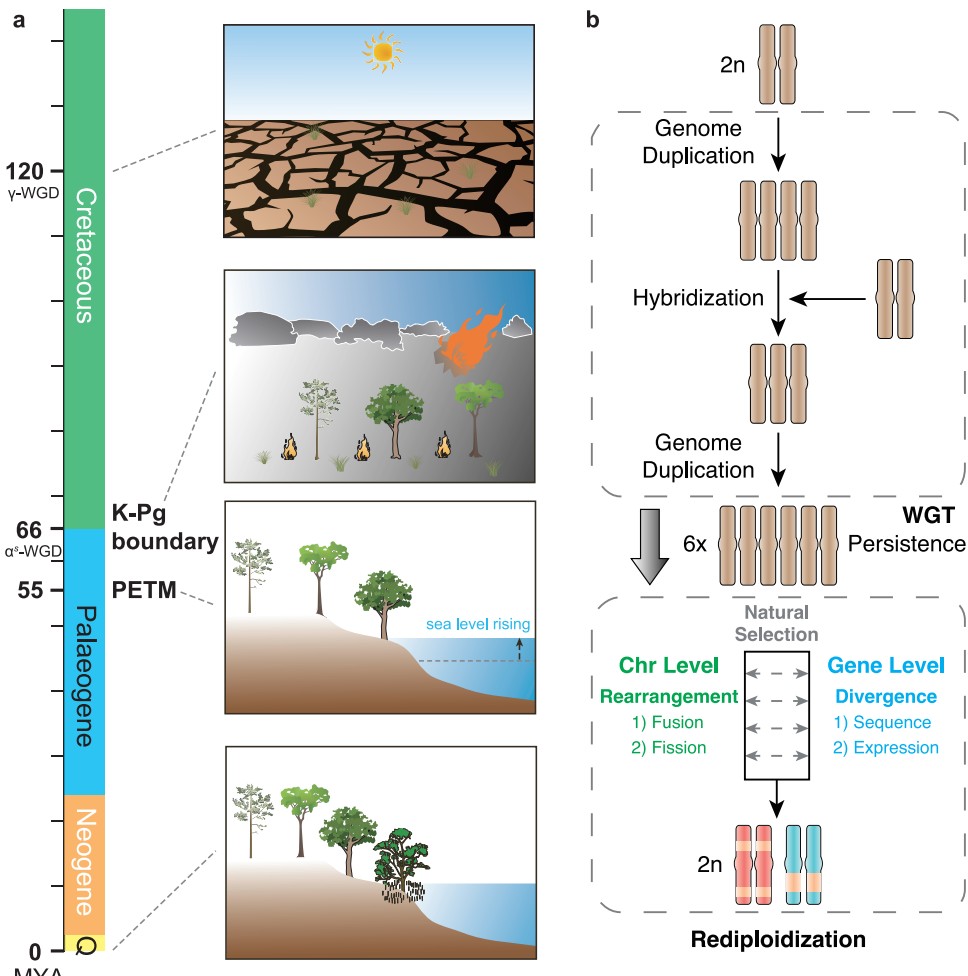

**Fig. 3 | A model of the polyploidization–rediploidization process in plants during global climate change. a** Whole-genome triplication (γ-WGT and α$^S$-WGT) events coincide with dramatic global climate changes. The sea level rise, massive extinction, and WGT event might provide the opportunity for the offshore woody plants to survive, leading to the emergence of the mangrove. The images portray the prevalent environments during various periods. In the Early Cretaceous (around 120 Mya), an arid climate prevailed[47]. At the K-Pg boundary (around 66 Mya), the dramatic global climate change following a significant asteroid collision with Earth[48]. During the PETM (around 55 Mya), there was a notable global temperature increase and a rise in eustatic sea levels[72]. Finally, the image depicts the current environment. The cartoon elements have been sourced and modified from materials contributed by Christine Thurber, Dieter Tracey, Jane Hawkey, Jane Thomas, and Tracey Saxby (available in the IAN Image Library at

https://ian.umces.edu/media-library/) under a CC BY-SA 4.0 License. Detailed credits for these cartoon materials can be found in Supplementary Data 2. **b** A hypothetical model of polyploidization–rediploidization process. The initial diploid genome experiences whole-genome triplication around the period of dramatic global environment and climate change. Polyploidy may persist during this period. Rediploidization post polyploidization is a major process for polyploids, driving the genome toward a diploid state through divergence of homologous genes in terms of sequence and expression, redundancy reductions, and large chromosome rearrangements such as fusion and fission events. The round of polyploidization and rediploidization process is widespread in angiosperms. The brown chromosomes represent homologous chromosomes, while the red and blue chromosomes represent significantly diverged chromosomes. The yellow bands indicate regions derived from other ancestral chromosomes through chromosomal rearrangements.

## WGT retained genes for root development and salt tolerance

Mangrove species live in environments characterized by high salinity and waterlogging, which pose challenges to plant growth and productivity[94,95]. The special root systems and high salt tolerance observed in mangroves are particularly noteworthy. *Sonneratia alba*, a prevalent and salt-tolerant mangrove species found in low intertidal zones, has evolved specialized structures like pneumatophores to enhance its waterlogging tolerance[51]. Following the WGT event, duplicated genes are often rapidly lost, while retained duplicates potentially changing expression or acquiring new functions serve as important sources of evolutionary innovation and aid in survival within the newly acquired habitat[56,57,79,96]. Therefore, we conducted functional analyses among the retained genes, which encompassed GO enrichment (Supplementary Fig. 27 and Supplementary Note 2) and gene function assessments based on annotations. Our focus was particularly directed toward the 584 three-copy retention groups generated by the

WGT event. Several gene groups involved in auxin distribution regulation, auxin signal transduction, reactive oxygen species (ROS) scavenging, ion transport, salt overly sensitive (SOS) signaling pathway, abscisic acid (ABA) signaling pathway, and transcriptional regulation retained the three duplicates ultimately (Supplementary Data 4). Interestingly, genetic and physiological experiments have demonstrated that salt modulates root growth direction by causing asymmetric auxin distribution and impairing the gravity response. In response to high salt levels, the SOS signaling pathway mediates the rapid degradation of amyloplasts in root columella cells, leading to the loss of root gravitropism[97,98]. *PIN* and *ABCB* genes, coding transporters polarly localized at the plasma membrane, promote auxin efflux activity[99,100], while PP2A proteins also influence PIN localization and participate in the regulation of auxin distribution[101]. Peroxidase protein-coding genes (*POD*) play a role in reducing ROS level, thereby preventing ROS from catalyzing auxin oxidation. These mechanisms

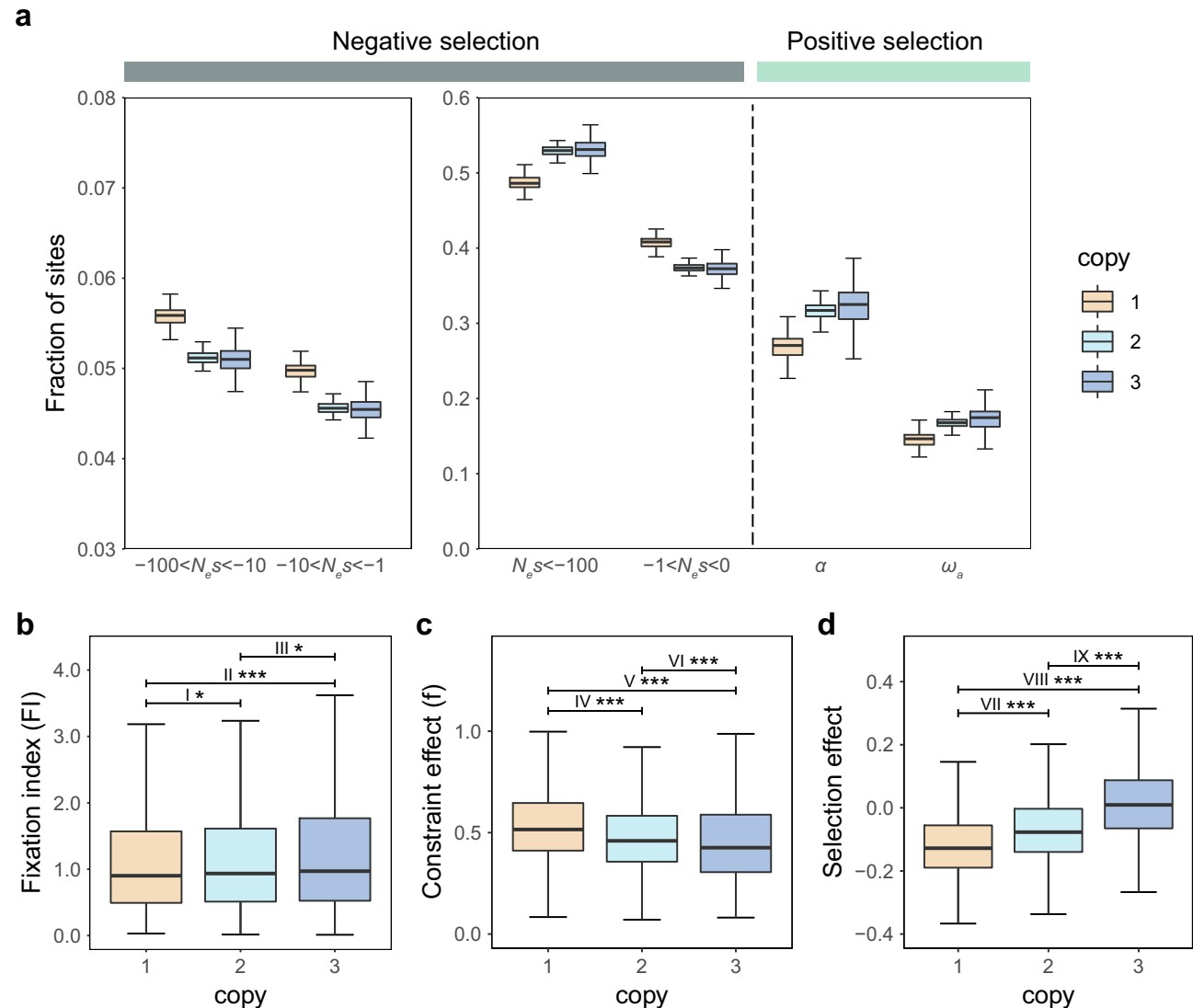

**Fig. 4 | Natural selection patterns among different-copy retention groups.**
**a** The inferred distribution of fitness effects ($N_es$), proportion of adaptive divergence ($\alpha$), and rate of adaptive substitution relative to neutral divergence ($\omega_a$) for different-copy retention groups generated by the WGT event. Each distribution was estimated based on 200 bootstrap replicates. Traditional MK test with fixation index (**b**), constraint effect (**c**), and selection effect (**d**) for these different-copy retention groups. The two-tailed $t$ test was applied to test for all pairwise differences. *P*-values are indicated by single asterisks (*P*-value < 0.05) or triple asterisks

(*P*-value < 0.001). I* (*P*-value = 3.9 × 10⁻²), II*** (*P*-value = 4.5 × 10⁻⁴), III* (*P*-value = 1.6 × 10⁻²), VI*** (*P*-value = 8.3 × 10⁻⁴), IV***, V***, VII***, VIII***, and IX*** (*P*-value < 1 × 10⁻¹⁵) represent the significantly different values for pairwise comparisons between groups. Box edges indicate upper and lower quartiles, centerlines indicate median values, and whiskers extend to 1.5 times the interquartile range. The number of genes in each retention group (one-copy, two-copy, and three-copy) was 3439, 4832, and 1171, respectively. Source data are provided as a Source Data file.

likely together facilitate the development of erect lateral branches in horizontal roots and shape the pneumatophores of *S. alba*, enhancing its waterlogging tolerance (Fig. 5).

Furthermore, we integrated transcriptomes by salt gradient experimental treatments to elucidate the mechanism underlying salt tolerance in *S. alba*. Using the HISAT2-HTseq-DESeq2 workflow, we examined expression profiles and identified differentially expressed genes (Supplementary Fig. 24). We observed 83 three-copy retention groups and 283 two-copy retention pairs with at least one copy showing up-regulation in leaf or root tissues under high salt conditions (Supplementary Table 9). We noticed that a subset of these genes, particularly the three-copy retentions, were associated with the phytohormone abscisic acid (ABA) (Fig. 5), including the ABA pathway, ABA transport, and other related processes (Supplementary Fig. 28). In detail, the release of Ca²⁺ in response to high salt triggers ABA biosynthesis[102]. Proteins such as PP2C and ABI5 function in the core

ABA signaling pathway and regulate downstream stress response genes, including *late embryogenesis abundant* (*LEA*)[103]. The expression of two *LEA* genes, *SalO09147* and *SalO11573*, was found to increase across salinity in leaf tissue. These hydrophilic and heat-stable proteins, with biased amino acid compositions, can sequester accumulated ions within cells and act as chaperones to prevent protein aggregation and inactivation[104,105]. Transcription factors have the potential to regulate multiple aspects of salt adaptation, with MYB and ERF positively influencing genes such as *ABI5* and *LEA*, and MYB2 and MYC2 acting as transcriptional activators in ABA-inducible gene expression[106]. Thus, the up-regulated retained genes can enhance plant desiccation and salt tolerance, contributing to adaptation in intertidal zones.

In summary, we successfully constructed chromosome-scale genomes for two Lythraceae plants, *S. alba* and *L. speciosa*, by combining PacBio SMRT sequencing, short reads sequencing, and Hi-C

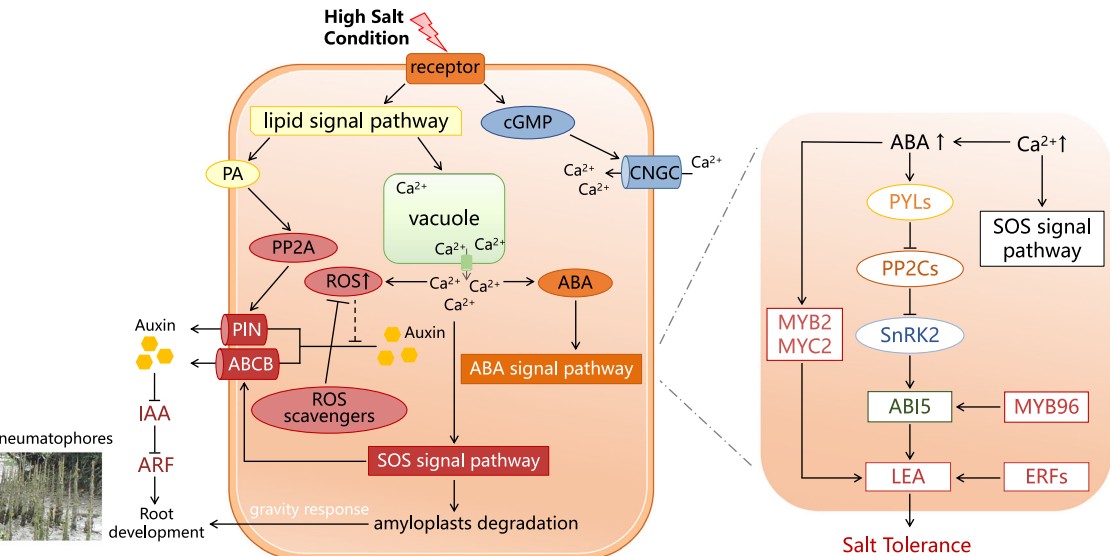

**Fig. 5 | The pathway of root development and salt tolerance in *S. alba*.** Salt modulates root growth direction by causing asymmetric auxin distribution and reducing the gravity response. Several genes may together facilitate the erect lateral branches of the horizontal roots and shape the pneumatophores. The duplicates with at least one copy up-regulated under high salt conditions, related to salt tolerance, are shown on the right. These genes are WGT retained duplicates and detailed three-copy retention groups are listed in Supplementary Data 4.

technologies. Based on genomic evidence and the previous studies[35,58,59,77,84,85], we report an improved model of the polyploidization–rediploidization process in plants, shedding light on adaptive evolution during periods of global climate change. Our findings revealed that *S. alba* and *L. speciosa* underwent a WGT event at approximately 64 Mya, which coincided with the K-Pg boundary. Subsequently, the mangrove tree experienced extensive chromosomal rearrangements and fractionations, leading to its modern genome structure. We further discovered that the retained duplicates from the WGT event in *S. alba* exhibited not only sequence divergence but also significant expression divergence, which is a crucial mechanism for rediploidization. Overall, our study contributes valuable insights into the plant evolution.

## Methods

### Plant materials

We sampled mature specimens of *Sonneratia alba* (Supplementary Fig. 2) and *Lagerstroemia speciosa* (Supplementary Fig. 3) from the nursery of Dongzhai Harbor National Nature Reserve in Haikou and Sun Yat-sen University in Guangzhou with proper permission, respectively. Fresh and healthy tissues were carefully collected and rapidly frozen in liquid nitrogen. Subsequently, the samples were stored at −80 °C in the laboratory until DNA or RNA extraction was performed.

### Library construction and sequencing

High-molecular-weight (HMW) genomic DNA was isolated from *L. speciosa* leaf tissue using the CTAB (hexadecyltrimethylammonium bromide) method[107] for both PacBio Single-Molecule Real-Time (SMRT) long-read sequencing and Illumina short-read sequencing. A PacBio SMRT-bell library was prepared with 10 kb long inserts following the manufacturer's protocol and subsequently sequenced on a PacBio Sequel II platform (Pacific Biosciences, Menlo Park, CA, USA). The generated PacBio reads underwent data filtering and preprocessing, resulting in 9.57 million reads, corresponding to approximately 95.60 Gb of data and ~299X coverage (assuming a genome size of 320 Mb, Supplementary Table 1). The same batch of genomic DNA was fragmented using sonication to construct a short-insert paired-end library with 500 bp inserts. This library was sequenced on an Illumina

HiSeq X Ten platform (San Diego, CA, USA), producing 41.16 Gb of data (Supplementary Table 1).

To facilitate gene prediction, total RNA was extracted from leaves of *L. speciosa* using the TRIzol universal reagent (Invitrogen) according to the manufacturer's instructions. The resulting RNA-seq library was sequenced on an Illumina HiSeq X Ten platform (San Diego, CA, USA). Furthermore, total RNA for the expression atlas of *S. alba* was extracted from leaf, root, flower, and fruit tissues of mature plants in the Dongzhai Harbor National Nature Reserve nursery. Each tissue contains three independent biological replicates. RNA-seq libraries were prepared for sequencing on an Illumina HiSeq 2500 platform (San Diego, CA, USA), generating 150 bp paired-end reads. The RNA-seq reads yielded a total of 76.76 Gb of data (Supplementary Table 5).

For Hi-C library construction, tender leaves of both *S. alba* and *L. speciosa* were subjected to formaldehyde fixation and subsequent lysis. The cross-linked DNA was digested with MboI, and the resulting restriction fragment ends were biotinylated and ligated. The purified DNA was then physically sheared to an approximate length of 400 bp. The Hi-C library of *L. speciosa* was sequenced on an Illumina NovaSeq 6000 platform (San Diego, CA, USA), while the Hi-C library of *S. alba* was sequenced on a BGISEQ-500 platform (Shenzhen, China).

### Genome assembly

We reported the genome assembly of *L. speciosa* and improved the previous assembly of *S. alba*[54]. The genome size of *S. alba* was estimated to be 211.67 Mb (Supplementary Fig. 29) through k-mer-based analysis[108]. The genome size of *L. speciosa* was initially estimated using flow cytometry and k-mer-based analysis. The flow cytometry measurement indicated a size of 361 Mb, consistent with the k-mer-based estimation of 340.46 Mb (Supplementary Fig. 30). Then we assembled the de novo genome of *L. speciosa* based on the PacBio long reads using wtdbg2[109] with optimized parameters. To improve the accuracy of the primary assembly, assemblies were further polished with Quiver (SMRT Analysis v2.3.0)[110] using long reads. We further removed residual errors using pilon (v1.22) based on Illumina paired-end reads[111].

Based on Hi-C data and primary genome assemblies of *S. alba* and *L. speciosa* (Supplementary Table 10), we improved them to generate pseudo-chromosome scale genomes, respectively. The Hi-C data underwent quality evaluation and assessment using HiC-Pro[112].

Subsequently, the Hi-C maps were generated using Juicer[113], and the scaffolds were roughly separated using Juicebox[114]. Manual corrections were made to resolve any misassemblies based on the observed interactions. The validated assemblies were then utilized to construct pseudo-chromosomes using the 3D-DNA tool[115]. These pseudo-chromosomes provided a chromosome-scale representation of the genomes, enhancing their structural organization and contiguity.

## Genome annotations

We identified repetitive sequences in each of the whole genomes using a combination of homology-based and de novo approaches. Initially, known TEs within the genome were identified using RepeatMasker with the Repbase TE library, and RepeatProteinMask searches against the TE protein database were conducted. Subsequently, a de novo repeat library for each genome was constructed using RepeatModeler, allowing for comprehensive analysis, refinement, and classification of consensus models for potential interspersed repeats[116]. Additionally, a de novo search for long terminal repeat (LTR) retrotransposons in each genome sequence was performed using LTR_FINDER (v1.0.7)[117]. Tandem repeats were identified using the Tandem Repeat Finder program, while non-interspersed repeat sequences were detected using RepeatMasker. The results from both approaches were integrated, and RepeatMasker was employed to identify the repeat sequences. We also estimated the age structures of long terminal repeat-retrotransposons (LTR-RTs). After de novo prediction of LTR-RTs, we imposed the criterion that an intact LTR-RT must be separated by 1 to 15 kb from other candidates, flanked by a pair of putative LTRs ranging from 100 bp to 3000 bp, with a similarity of over 80%, and possessed a complete *Gag-Pol* sequence. The timing of LTR-RT insertion was estimated based on the divergence between the 5′-LTR and 3′-LTR of the same transposon, using a mutation rate of $1.3 \times 10^{-8}$ substitutions per year per site[63].

We conducted gene model prediction of each genome using a combination of ab initio, homology-based, and RNA-seq-assisted prediction. We used Augustus (v3.3.1)[118] and GeneMark[119] to perform ab initio gene prediction based on the masked genome except for the low complexity or simple repeats, because some of these repeats could be found in the genes. The protein sets were collected for homology-based prediction and chosen as homology-based evidence from sequenced relative plant species and model plant species. Then exonerate (v2.2.0) was used to generate the gene structures based on the homology alignments[120]. Clean RNA-seq reads were mapped against the assembly using Tophat2 (v2.1.1), and transcripts were identified using Cufflinks (v2.2.1)[121,122]. Finally, EvidenceModeler was used to integrate all predictions to generate non-redundant and consensus gene models[123]. Gene functions were annotated based on the best alignment matches to the NCBI (NR), Swissprot, TrEMBL, InterPro, the Kyoto Encyclopedia of Genes and Genomes (KEGG), Gene Ontology (GO), and Pfam non-redundant protein databases. The transcription factor identification was performed using iTAK (v1.7)[124]. The quality of genome assembly and annotation was assessed using Benchmarking Universal Single-Copy Orthologs (BUSCOv5) with the plant-specific dataset (eudicotyledons_odb10)[125].

## Phylogeny reconstruction and molecular dating

We downloaded genome and annotation data for *Vitis vinifera* (Genoscope.12X)[73], *Arabidopsis thaliana* (TAIR10)[126], *Eucalyptus grandis* (v2.0) from Phytozome v12.1 database[83]; *Nelumbo nucifera* from Nelumbo Genome Database[127]; *Punica granatum* (GCF_007655135.1) from NCBI database[128]. We used OrthoFinder to identify orthologous genes among *S. alba*, *L. speciosa*, and these five eudicot species, resulting in the identification of single-copy gene groups[129]. For each group, we aligned the corresponding single-copy orthologous proteins and generated codon alignments using MAFFT[130] and PAL2NAL[131]. To ensure data quality, we further applied Gblocks

0.91b[132] to trim the alignments and discarded ambiguity alignments shorter than 150 bp. Using the concatenated alignment of these groups, we inferred a phylogenetic tree using RAxML-NG[133] with the GTR + GAMMA + I model and performed 1,000 bootstrap replicates. Following its reconstruction, we estimated the divergence time among the seven species using MCMCTREE from the PAML (v4.9j) package with approximate likelihood calculation[134,135]. The HKY85 + G nucleotide substitution model and independent-rates clock model were employed in the molecular dating. To provide calibration points, we incorporated two reliable fossil calibrations. Firstly, the root node of eudicots was placed at 119.6–128.63 Mya[136]. Secondly, the common ancestor of *Sonneratia* and *Lagerstroemia* was set to a time earlier than 55.8 Mya, since the earliest convincing fossils of *Sonneratia*-like pollen[137]. In order to delineate the positions of these plants within Lythraceae, we expanded our analysis by constructing a more extensive phylogenetic tree using these seven plants, other 35 genome-sequenced angiosperms, and the gymnosperm *Gnetum montanum* as an outgroup (Supplementary Data 5). Utilizing the embryophyta_odb10 lineage ancestral variant dataset (comprising a consensus sequence and variants of extant sequences) in BUSCOv5[125], we identified 868 low-copy nuclear genes. We then performed sequence alignment and phylogenetic inference as described earlier. The early divergence times in angiosperms were set to 125–247.2 Mya[138,139]. All MCMC analyses were independently run twice to ensure convergence, with 10 million generations and sampling every 500 generations after a burn-in of 1,000,000 iterations. The phylogenetic trees were visualized using the R package GGTREE[140].

## Whole-genome triplication analyses

In order to identify and locate putative WGDs in Lythraceae species, we used a multipronged approach, including the intra- and inter-species synteny analysis, Ks-base estimation, and phylogenetic reconciliation. Initially, we utilized the BLASTP program to align protein sequences between species (*P. granatum* vs. *S. alba*, *P. granatum* vs. *L. speciosa*) and within species, applying the parameters (identity ≥30%, e-value < 1e−10, alignment length ≥30% of both query and reference sequences). We identified syntenic blocks containing a minimum of five shared genes using MCScanX[141], and the resulting syntenic blocks between species were visualized by Circos[142]. Subsequently, we applied KaKs_Calculator to calculate synonymous substitution rates (Ks) with the YN substitution model[143] based on alignments of all syntenic gene pairs and constructed Ks distribution. To identify paralogous genes generated from the WGD event in the *S. alba* genome, we selected blocks with median Ks values in the range of 0.2-1.0, excluding gene pairs with Ks values larger than 1.26. Using the R package igraph (https://igraph.org), we further classified different-copy retention groups after the WGD event (Supplementary Fig. 12). We also identified different-copy retention groups after the WGD event in *L. speciosa* using the same workflow (Supplementary Fig. 13). The analyses of synteny and Ks-base indicated that both *S. alba* and *L. speciosa* had undergone a whole-genome triplication (WGT) event. Simultaneously, we illustrated the distribution of gene densities for different-copy retention groups in both species (Supplementary Fig. 14). To investigate whether the WGT event was shared between *S. alba* and *L. speciosa*, we identified 306 *P. granatum* genes that possessed three orthologs generated by the WGT in both *S. alba* and *L. speciosa*. Then we performed gene tree reconstruction using RAxML-NG and classified the phylogenetic trees based on their topologies.

We also inferred and located the putative WGT placement using the multi-taxon paleopolyploidy search (MAPS) tool[144]. Clustering gene families among five species, including *S. alba*, *L. speciosa*, *P. granatum*, *E. grandis*, and *A. thaliana*, by OrthoFinder, we retained the gene families with at least one gene present in each species. We constructed gene trees based on multiple sequence alignments of each

gene family as described above and rooted each tree using Notung (v2.9.1.5)[145]. By mapping these gene trees to the given species tree, we calculated the percentage of subtrees with gene duplications shared by all species descended from each node using the MAPS tool. To validate the WGT placement, we compared the subtree percentages at each node among observed, null simulated, and positive simulated data and recognized a significant gene duplication burst indicative of a polyploidy event. Background gene birth and death rates were estimated using the R package WGDgc[146] and the mean of a geometric distribution of the root was calculated through CAFE analysis[147]. We performed 2,000 simulated gene tree simulations with 200 bootstrap replicates for both null and positive simulations. In the positive simulation, we designated a polyploidy event in the common ancestor of *S. alba* and *L. speciosa*, setting the wgd_retention_rate to 0.2. The observed and simulated data were compared to evaluate the location of the WGT events. The results of the synteny, Ks-based, and phylogenetic analyses consistently indicated that *S. alba* and *L. speciosa* had undergone a whole-genome triplication (WGT) event prior to their diverging from a common ancestor.

To determine the absolute timing of the whole-genome duplication (WGT) event, we conducted a molecular clock analysis on concatenated gene families, calibrated using species divergence times[148]. Specifically, we first identified 208 homologous gene groups among *S. alba*, *L. speciosa*, and *P. granatum* supporting the WGT event before the speciation event (Fig. 2e). For each group, we used the reciprocal BLASTP best-hit method between *P. granatum* and *E. grandis* to obtain the corresponding ortholog from *E. grandis*. Ultimately, we identified 170 gene families that exhibited a clear signal of the WGT event on the common ancestor of *S. alba* and *L. speciosa*. In each gene tree, the genes from *S. alba* and *L. speciosa* were divided into three clades. To improve the robustness and precision of estimation, we selected the outgroup clade and one of the two ingroup clades randomly to concatenate multiple sequence alignments. The phylogenetic tree was then constructed using RAxML-NG, and the molecular clock analysis was performed using the approximate likelihood calculation method in MCMCTREE under the appropriate model[134,135]. The nodes were constrained using species divergence times obtained from the phylogenetic tree as described above. Each analysis was independently run twice to ensure convergence.

## Chromosome evolution

In order to infer their evolutionary history, we selected representative species in the Lythraceae with chromosome-scale genome assemblies. We first inferred the ancestral chromosome numbers across the phylogenetic tree (Supplementary Fig. 18) using ChromEvol (v2.1)[149]. The haploid chromosome number ($n$) of *Sonneratia alba* was reported by S. Graham[150], while the $n$ of other Lythraceae species (*Lagerstroemia speciosa*, *Punica granatum*, *Pemphis acidula*) and outgroup (*Eucalyptus grandis*) were obtained from the Chromosome Counts Database (CCDB)[151]. The chromosome number of the most recent common ancestor (MRCA) among *S. alba*, *L. speciosa*, and *P. granatum* was the same as *P. granatum* ($n = 8$). We utilized WGDI (v0.6.5) to identify adjacent conserved collinear genes and blocks among all chromosome pairs within the three Lythraceae species, and then reconstructed the Ancestral Lythraceae Karyotype (ALK), excluding interference from fragmented collinear regions, following the tutorial[152,153]. Subsequently, we visualized the global pattern of chromosomal changes in extant species. Furthermore, we depicted the evolutionary history of *S. alba* chromosomes to provide a clearer representation of the karyotype evolution[152]. While the reconstructed ancestral karyotype almost certainly has a very similar structure to the true ancestral genome, it may not be absolutely identical[60]. We also conducted synteny analysis among the modern genomes of the three Lythraceae species using MCScanX and JCVI to discover chromosome rearrangements[154].

## Transcriptome sequencing and analysis

RNA-seq reads from the leaf, root, flower, and fruit tissues of *S. alba* were first filtered using SolexaQA++ (v3.1.7.1)[155]. Clean reads were aligned to the *S. alba* genome using HISAT2 (v2.2.0)[156]. The HTSeq (v0.13.5) was utilized to determine the number of reads uniquely mapped to each gene in the tissue samples[157]. To detect the differential expression of duplicated genes, we employed the exact conditional test[158], which has been successfully applied in soybean, *Brassica*, and *Avicennia*[159–161]. For each pair of duplicated genes, we computed the *P*-value using the R function *binom.test*. Multiple testing was corrected by applying the Bonferroni correction method. Differential expression was considered significant for gene pairs with a corrected *P*-value below 5%. Only gene pairs whose at least one gene read number more than 0 were included in the analysis. We also applied the method for three-copy duplicated gene groups through pairwise comparisons. We finally identified duplicated gene pairs with differential expression for each tissue based on the consistency in three replicates. To investigate the functional roles of these differentially expressed gene pairs, we performed GO enrichment analysis using BiNGO in Cytoscape (v3.7.2)[162]. Additionally, we performed RNA-seq on leaf, stem, flower, and fruit tissues of *L. speciosa* (Supplementary Table 7 and Supplementary Fig. 31). We employed the HISAT2–HTSeq–exact conditional test workflow, as described earlier, to identify differentially expressed duplicated gene pairs. Subsequently, we conducted GO enrichment analysis on these gene pairs in *L. speciosa* (Supplementary Fig. 32).

We also collected the transcriptomes of leaf and root tissues of *S. alba* under different salinity conditions[163]. Specifically, *S. alba* plants were divided into three groups and subjected to irrigation with solutions containing 0, 250, and 500 mM NaCl to simulate low, medium, and high salinity conditions, respectively. After reads mapping and counting, we identified differentially expressed genes (DEGs) from two comparisons (medium *vs.* low salinity condition, high *vs.* medium salinity condition) using the R package DESeq2[164]. The *P*-value below 5% and fold change greater than two was set as the significantly differential expression threshold.

## Distribution of fitness effects of variants

To quantify the impact of selection on different-copy retention groups after the WGT event, we estimated the distribution of fitness effects (DFE) of new mutations and the proportion of adaptive divergence ($\alpha$) at zero-fold nonsynonymous sites using DFE-alpha[88–90]. The whole-genome resequencing data from two populations (Cebu, Philippines; Davao, Philippines) of *Sonneratia alba* (12 individuals per population), and one individual of congeneric species *S. apetala* were used in the analysis. We first binned protein-coding genes into three subsets according to retained copy numbers (one-, two-, three-copy). We used DFE-alpha to compare the folded site frequency spectrum (SFS) and divergence of zero-fold nonsynonymous sites with those for four-fold synonymous sites. The four-fold synonymous sites were assumed to be neutral. We also estimated DFE and $\alpha$ with 200 bootstrap replicates generated by randomly sampling genes of each subset.

## Reporting summary

Further information on research design is available in the Nature Portfolio Reporting Summary linked to this article.

## Data availability

The raw genomic Illumina reads, PacBio reads, Hi-C reads, and RNA-seq reads reported in this paper have been deposited in the Genome Sequence Archive (GSA, https://ngdc.cncb.ac.cn/gsa) in National Genomics Data Center, Beijing Institute of Genomics, Chinese Academy of Sciences / China National Center for Bioinformation, under accession number CRA004284 with BioProject ID PRJCA005319. The genome assembly sequences have been deposited in the Genome Warehouse (GWH, https://ngdc.cncb.ac.cn/gwh) in National Genomics Data Center

under accession number GWHBCIQ00000000 [https://ngdc.cncb. ac.cn/gwh/Assembly/20653/show], GWHBCKL00000000 [https:// ngdc.cncb.ac.cn/gwh/Assembly/20692/show] with BioProject ID PRJCA004930 and BioSample ID SAMC353197, SAMC353201. The genome assemblies and annotations are also available at Figshare: *Sonneratia alba* [https://doi.org/10.6084/m9.figshare.25118819], *Lagerstroemia speciosa* [https://doi.org/10.6084/m9.figshare.25118831]. Source data are provided with this paper.

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

## Acknowledgements
The authors thank Zuyao Liu and Nan Wang for their technical support. The project was supported by the National Natural Science Foundation of China (32170230 to Z.H., 31971540 to Z.H. and 32330005 to S. Shi), the Guangdong Basic and Applied Basic Research Foundation (2023B1515020083 to Z.H.), the Innovation Group Project of Southern Marine Science and Engineering Guangdong Laboratory (Zhuhai) (311021006 to S. Shi), the Shenzhen Science and Technology Innovation Program (RCBS20221008093316043 to Q.C.), and the China Postdoctoral Science Foundation (2023M740690 to X.F.).

## Author contributions
Z.H. and S. Shi conceived the study. X.F., S. Shi, and Z.H. designed and conceptualized the study. X.F., Q.C., W.W., J.W., G.L., and Z.H. performed the data analysis. X.F., Q.C., G.L., S.X., C.Z., S. Shi, and Z.H. collected materials. X.F., Q.C., S.X., S. Shao, and M.L. performed the experiments. X.F., Q.C., S. Shi, and Z.H. wrote the manuscript. C.-I.W. revised the manuscript. All authors read and approved the final manuscript.

## Competing interests
The authors declare no competing interests.
