## [Peer Review File · Nature Communications]

Genomic evidence for rediploidization and adaptive evolution following the whole-genome triplicationReviewers' Comments:

Reviewer #1:

Remarks to the Author:

This study assembled the chromosome-scale genomes of two Lythraceae plants and effectively presents the process of rediploidization after recent whole-genome triplication and its potential role in adaptive evolution. The authors also propose a model of the polyploidization-rediploidization process in plants, which holds significance in understanding adaptive evolution during periods of global climate change. Furthermore, mangroves comprise an exceptionally captivating group of plants, thriving in distinct and specialized environments. *Sonneratia alba* is one of the most representative mangrove species. Employing this clade as a case study, the authors have performed comprehensive analyses. These analyses yield a lot of genomic evidence about rediploidization in different aspects, such as sequence and expression divergence, chromosomal evolution, selective pressures, sub-functionalization, and adaptive trait evolution. Overall, I am glad to review this nice work and it is helpful to understand plant evolution. I am pleased to recommend it for publication after revisions.

More comments are listed below.

1. Line 52-53. The authors concisely outline several distinct WGT events within eudicots. Notably, there are sequenced hexaploid species among eudicot plants, such as *Solanum nigrum* (Lee et al., 2023) and *Chrysanthemum seticuspe* (Nakano et al., 2021). Please add these species to the existing tree and update Supplementary Figure 1.
2. If *S. alba* and *L. speciosa* have shared WGT event, then why the k_s peak is not the same. This need to be explained.
3. I also suggest to calculate the gene numbers between different species in HOGs obtained by orthofinder. If there were a WGT, the proportion of 1:3:3 must be much higher than 1:2:3 or something else.
4. Line 136-138, 154-155. The authors skillfully integrated a trio of methodologies encompassing synteny, K_s -base, and phylogenetic approaches to characterize the occurrence and location of the whole-genome triplication event in *S. alba* and *L. speciosa*. This strategy effectively addresses the challenge posed by varying evolutionary rates across different plant species, ensuring more robust identification of features and positions of polyploidy events. Please provide a flowchart to illustrate it and enhance clarity.
5. Line 182-194. The authors performed a chromosome-scale comparative investigation among mangrove species and its relative *L. speciosa* and found the mangrove genome experienced more chromosome changes. To improve the understanding of the distinct habitats, please provide additional information about the habitats of the related species.
6. Line 197-200. The authors mentioned, "During periods of dramatic global environment and climate change, newly formed polyploids typically possess a significant fitness advantage over diploids". In support of this assertion, they have illustrated the specific global environment when the WGD/WGT happened in Fig. 3a. However, these images are devoid of textual descriptions. Please provide descriptions with references that elucidate the environment prevalent during that period.
7. Line 196-211. The authors propose a model elucidating the polyploidization-rediploidization process. It seems that a cyclical nature in plant genome dynamics. It prompts consideration of the current stage of this cycle within plants. What stage are plants in now? Whether there are different stages between the mangrove species and its relatives? The authors need to think about and discuss it in this section.

8. Line 204-205. Please add the references to this point ("all angiosperms have undergone successive rounds of polyploidization and rediploidization process").

9. Line 257-296. By conducting transcriptome analysis, the authors identified WGT-retained genes related to adaptive evolution. In addition to the presentation of figures and tables, please provide a concise summary of these identified genes, along with their sequences. It could facilitate the utilization.

10. Line 400-401. What model was used in the MCMC analysis?

11. Line 505. The abbreviation of "confidence intervals" should be "CIs".

12. Fig. 2d looks like a partially collinear relationship. Kindly make a note of this observation in the figure legend.

13. Line 880. "Supplementary Table S7" should be "Supplementary Table 7".

14. Please add a row of anchored gene numbers to Table 1.

15. Please state the source of the pictures (a, b, c) in Supplementary Figure 3.

16. Please change "Hi-C" to "Hi-C reads" in Supplementary Table 1.

Reviewer #2:

Remarks to the Author:

This is a very interesting manuscript taking a novel paired rediploidisation system. It presents a unique look at the rediploidisation process over medium-scale (not recent) evolutionary time. It will be of significance to workers in several fields, from ecology and evolution to fundamental plant biology. In general Does the work supports the conclusions and claims and I have no major comments that require modification of the analysis; as far as I can ascertain, the methodology is sound. I provide more minor comments below with the intention of helping to enhance the presentation.

26: 'recent': 64 million years ago is not what most in the field would call a recent polyploidy event. Nor is it necessary for the novelty here to suggest this is recent, so I propose to strike all use of 'recent'.

32: 'adaptive value' can be inferred by retention, but 'accelerated evolutionary adaptation' cannot.

34: 'we propose a model': if you say you propose a model in the abstract, say something about it other than selling the quality of your genomes that served as input data. I also note that the model in the figure is a little bit 'light'; the novelty of the 'model' is to me not great, so I would suggest stressing the new model much, to be frank.

54-57: this gives the false impression that WGD is an adaptive panacea when in fact most young polyploids are not stable. This downside can be clearly seen in discussion in e.g. <https://doi.org/10.3389/fevo.2018.00117>

58: 'Plants on the Earth': suggest rephrase to 'Plants' or some other formulation; it's awkward as is.

66: 'rain forests of the sea' is puzzling to me and seems not relevant. Perhaps it's a commonly used term for these species, but seems to me a little bit misleading. Rainforests are much more species-diverse in my mind than mangroves.

78: references 51 and 52 are introduced as examples or support for the statement that high quality genomes have provided opportunity to study genomes post WGD. Immediately following this statement the current study is introduced, but there is no suggestion what the gap was following studies 51 and 52 or what the major current innovation motivating this study in relation to them. It

would be good to say here how the current study contributes something truly novel.

83-84: 'The genome sequences... evolutionary research in plants' is far too weak a statement to make any interest. Suggest rewriting the end of the intro to be more powerful, stating the specific point of this study, as it currently is very weak indeed.

88: it is strange to start here with HiC. What about the sequencing of the contig assembly? There is no reference given for this, so where do I go to learn about how it was originally sequenced?

90: '204.46 Mb': what was the expected size based on flow or kmers? I see these data are the methods; it would be better if they were in the results here too/instead.

96: 'consistent with the estimated genome size': what is that estimate?

100: a sentence about how the annotation was done would be nice in the results.

107: suggest to rephrase to either 'Fewer TEs accumulate' or 'Less TE accumulation'

123: 'taxa' is plural; 'taxon' is singular, which is what I think you want here.

144, 5, 7, 152, 171: 'recent': this happened at around the time of the K-Pg boundary! I would disagree with the use of the word 'recent', and besides it does no benefit to call this WGT 'recent' for your story. You don't need it.

197-8: There is evidence of increased signal of WGD at times of dramatic environmental and climate change, but there is absolutely not sufficient evidence to make such a strong statement that 'newly formed polyploids typically possess a significant fitness advantage over diploids.' Typically neopolyploids are dead due e.g. in autotetraploids to meiotic catastrophe. I am passionate about polyploids and have devoted my career to studying them and I therefore also want to say this, but this statement goes much farther than the data and must be revised. See a more balanced discussion of this in e.g. the review I reference in 54-57 above. One much more appropriate suggestion would be to say: "newly formed polyploids can possess fitness advantages over diploids" or more accurately something like "signals of WGD persistence correlate with times of environmental and climate change, suggesting potential benefit to WGD in the face of challenges".

201: 'will': here too the language is too simplistic. I would suggest 'may' and to again see the review I reference above for more balanced examples.

197-202: the studies referenced here are not strong and often rather old. Again, the statements should be more moderate and less absolute and referencing various excellent reviews of this broad literature in addition to the one I suggest would be much more informative: e.g. doi.org/10.1086/700636 and [doi: 10.1101/gad.271072.115](https://doi.org/10.1101/gad.271072.115).

265: 'functional analysis': do you mean GO analysis? This is unclear in the text. Describing this analysis better in the main text is needed. The results seem interesting, but they need to be better shown. Nor do I see a clear description of this analysis in the methods.

Between the standard GO analysis and the 'functional analysis' not well described, it seems the enriched terms are rather 'high level' and not very descriptive: e.g. 'regulation of biosynthetic process', 'regulation of metabolic processes', 'biological regulation'. Can the authors better highlight the more specific GOs that are better describing particular processes or functions?

268-9: these are of course very interesting pathways!

298: no need to say 'highly accurate'

The conclusion is much better without the last sentence to be honest. There is no need at all for such a broad statement, which reduces the impact of the rest of the nice writing.

319: more details should be given on the CTAB and reference it.

Reviewer #3:

Remarks to the Author:

The authors sequenced and assembled genomes of two related species from the family Lythraceae. One species belongs to mangrove species, while the second species (*Lagerstroemia speciosa*) is not adapted to the mangrove ecosystem. Overall the paper is clearly structured and well-written, however I need to review this work in comparison with similar studies published in Nat Comm and similar journals recently. In this comparison I found the present paper to contain less novelty than some other genome-based papers published in Nat Comm. Here are some key points I considered as critical:

(1) Page 3. „almost all mangrove species are currently considered diploids“. Please make sure that the current interpretation is correct. I did not check the cited references, however very often species are considered diploid due to the lack of other (lower) chromosome numbers in a given taxon (e.g., genus), and thus, species of *Helianthus* (n=17) are or were considered to be diploid (and there are more examples of course).

(2) Page 4, line 109... Here the authors are reporting the difference in the proportion of TEs between the two genomes, not really appreciating the fact that logically 12 centromeres will have much less repeats than 24 centromeres (*Lagerstroemia*). And so it is also logic to expect that the mangrove species will have smaller nuclear genome than the inland species, as it was found. This part is overall trivial and concluding that plants and the investigating species have LTR retrotransposons. Where are other types of repeats, including tandem repeats? The concluding sentence is somewhat naive but logic – see above – the authors should, for example, consider that logically there are less insertions of LTR-RTs in the mangrove genome due to the lower number of genomic regions where the insertions can be tolerated (typically these are pericentromere regions: 12 versus 24).

(3) Page 5. The phylogeny. With so many genome sequenced and analyzed phylogenetically I cannot be really happy about the phylogeny including 7 species (out of 250 000 as the authors correctly report in the Introduction)! It is also unclear and not discussed if their divergence time estimates are congruent with already published datings.

(4) The same page, line 144. „219 syntenic block pairs comprising 3,333 gene pairs in *P. granatum*“. And later: „suggesting that *P. granatum* did not experience recent polyploidy events.“ It remains to be explain where from the 219 syntenic block pairs in *P. granatum* come from.

(5) The usage of „recent“ for the identified whole-genome triplication 64 mya is really funny and I did not understand why the authors actually use „recent“ for this very old polyploidization event.

(6) Page 6. Line 156 and following. I really did not understand where from the authors get information how the WGT helped to cope with the K-Pg global extinction and later events? If I accept this tale, then it comes even before adaptive genes are introduced, researched and discussed.

(7) The same page: „If we consider only polyploidy, the haploid chromosome number would be 168 (21*2*2*2).“ It is not clear whether this is related to *Utricularia* or *Lythraceae* and, second, it has to be explained where 21 comes from (I guess it could be 3 x 7 of the Gamma WGT?).

(8) Page 7, 1st paragraph. To me, the fact that the ancestral genome had 8 chromosome pairs is repeated three times. This part is very vague and general – the reader can look at figure 2f but what can be seen and inferred from this karyograms? It is also not very clear that actually the inland genome has retained the ancestral number of chromosomes after the WGT, but there are chromosomal rearrangements (what types of chr. rearrangements?). Suppl fig 13: why *Pemphis acidula* with the duplicated ancestral genome is not mentioned and used for comparisons with *P.*

granatum and the two species investigated in here?

(9) I missed more indepth analysis how the hexaploid genome was rearranged in the two species investigated.

(10) The polyploidization-diploidization cycle (Figure 3) was published by several authors (some cited, some not) and, forgive me, to call this simple and already several times published scheme as A MODEL is little too much. The authors should eliminated „this model“ or modify it by acknowledging researchers who published basically the same cycle shemes earlier.

(11) Expression analysis. I was really not sure how this was meant and what exactly can be concluded when the same analysis was not done for the inland species and/or another (non-)mangrove species for which comparabile dataset is available.

(12) Also, surprisingly, I did not get any information how the hexaploid genome has been formed and if post-polyploid gene fractionation (diploidization process in general) impacted all three subgenomes in the same or different way.

Replies to reviewers' comments point by point:

Reviewer #1 (Remarks to the Author):

This study assembled the chromosome-scale genomes of two Lythraceae plants and effectively presents the process of rediploidization after recent whole-genome triplication and its potential role in adaptive evolution. The authors also propose a model of the polyploidization-rediploidization process in plants, which holds significance in understanding adaptive evolution during periods of global climate change. Furthermore, mangroves comprise an exceptionally captivating group of plants, thriving in distinct and specialized environments. *Sonneratia alba* is one of the most representative mangrove species. Employing this clade as a case study, the authors have performed comprehensive analyses. These analyses yield a lot of genomic evidence about rediploidization in different aspects, such as sequence and expression divergence, chromosomal evolution, selective pressures, sub-functionalization, and adaptive trait evolution. Overall, I am glad to review this nice work and it is helpful to understand plant evolution. I am pleased to recommend it for publication after revisions.

[Reply]: Thanks for the positive comments.

More comments are listed below.

1. Line 52-53. The authors concisely outline several distinct WGT events within eudicots. Notably, there are sequenced hexaploid species among eudicot plants, such as *Solanum nigrum* (Lee et al., 2023) and *Chrysanthemum seticuspe* (Nakano et al., 2021). Please add these species to the existing tree and update Supplementary Figure 1.

[Reply and Revision]: We have updated Supplementary Figure 1 by incorporating three hexaploid species (*Solanum nigrum*, *Chrysanthemum seticuspe*, and *Helianthus tuberosus*) and annotating relevant WGT events.

2. If *S. alba* and *L. speciosa* have shared WGT event, then why the K_s peak is not the same. This need to be explained.

[Reply]: Thanks. We understand the reviewer's concern. Synonymous substitution rates (K_s) between paralogous genes can be influenced by divergent evolutionary or substitution rates in different plants. While the K_s peak appears slightly different between *S. alba* and *L. speciosa*, we have employed phylogenetic approaches to pinpoint more accurate positions of the polyploidy event. Our multipronged approach allows us to overcome the challenges posed by divergent evolutionary rates in different plants, enabling the identification of more precise features and positions of polyploidy events. The revision below is much more explicit.

[Revision]: (Line 155-156) While the Ks peak appears slightly different between *S. alba* and *L. speciosa*, we performed gene tree reconstructions of the syntenic gene groups ...

(Line 159-161) This multipronged approach allows us to overcome the challenges posed by divergent evolutionary rates in different plants, enabling the identification of more accurate features and positions of polyploidy events^{23,68-70}.

3. I also suggest to calculate the gene numbers between different species in HOGs obtained by orthofinder. If there were a WGT, the proportion of 1:3:3 must be much higher than 1:2:3 or something else.

[Reply]: Thank you for your suggestion. We have calculated the gene numbers within different species in homologous groups (HOGs). The number of HOG (pgr:lsp:sal = 1:3:3) is 315, which is relatively small compared to the HOG with a ratio of 1:2:2 (1889). As the WGTs occurred relatively long ago, following the subsequent rediploidization process, only a limited number of homologous groups retained three full copies, with 584 in *S. alba* and 943 in *L. speciosa*. Therefore, the identification of accurate features and positions of polyploidy events necessitates a combination of synteny, Ks-based, and phylogenetic approaches to uncover traces on the genome. It is important to note that the data from HOGs may not fully reflect the features of WGT.

4. Line 136-138, 154-155. The authors skillfully integrated a trio of methodologies encompassing synteny, Ks-base, and phylogenetic approaches to characterize the occurrence and location of the whole-genome triplication event in *S. alba* and *L. speciosa*. This strategy effectively addresses the challenge posed by varying evolutionary rates across different plant species, ensuring more robust identification of features and positions of polyploidy events. Please provide a flowchart to illustrate it and enhance clarity.

[Reply and Revision]: Thank the reviewer for the helpful reminder. We have added the flowchart in the supplementary information (Supplementary Fig. 11).

5. Line 182-194. The authors performed a chromosome-scale comparative investigation among mangrove species and its relative *L. speciosa* and found the mangrove genome experienced more chromosome changes. To improve the understanding of the distinct habitats, please provide additional information about the habitats of the related species.

[Reply]: Thanks. We have added additional information about the habitats of the mangrove tree *Sonneratia alba* and related inland plant *Lagerstroemia speciosa* in Supplementary Notes.

[Revision]: *Sonneratia alba* inhabits low intertidal zones of downstream estuarine systems and is one of the most pervasive and salt-tolerant mangrove species widespread in the Indo West Pacific (IWP) region. Evolving specialized structures such as pneumatophores, *S. alba* demonstrates its waterlogging and salt tolerance, particularly in low intertidal zones.

Lagerstroemia speciosa, the closely related inland woody plant, demonstrates adaptability across diverse habitats, including lowland rainforests, riparian areas, as well as urban and rural environments. The cultivation of *L. speciosa* in gardens and urban landscapes highlights its ornamental value and widespread popularity.

6. Line 197-200. The authors mentioned, “During periods of dramatic global environment and climate change, newly formed polyploids typically possess a significant fitness advantage over diploids”. In support of this assertion, they have illustrated the specific global environment when the WGD/WGT happened in Fig. 3a. However, these images are devoid of textual descriptions. Please provide descriptions with references that elucidate the environment prevalent during that period.

[Reply]: We appreciate the reviewer’s comment. We have provided a detailed description of the images in the legend of Fig. 3. The revision below is much more explicit.

[Revision]: (Line 958-963) The images portray the prevalent environments during various periods. In the Early Cretaceous (around 120 Mya), an arid climate prevailed⁴⁷. At the K-Pg boundary (around 66 Mya), the dramatic global climate change following a significant asteroid collision with Earth⁴⁸. During the PETM (around 55 Mya), there was a notable global temperature increase and a rise in eustatic sea levels⁷². Finally, the image depicts the current environment.

7. Line 196-211. The authors propose a model elucidating the polyploidization–rediploidization process. It seems that a cyclical nature in plant genome dynamics. It prompts consideration of the current stage of this cycle within plants. What stage are plants in now? Whether there are different stages between the mangrove species and its relatives? The authors need to think about and discuss it in this section.

[Reply]: Polyploidy events have significantly influenced the evolutionary history of angiosperms, but diploid plants currently predominate. The process of rediploidization following polyploidization is crucial for polyploids, ultimately leading to modern descendants as normal diploids cytogenetically, generating important genetic and taxonomic diversity. Considering the potential role of ploidy changes in genome evolution, we improve a model based on new genomic evidence and the previous studies. This model explains the polyploidization–rediploidization process, elucidating the adaptive evolution during global upheavals and restoration.

While *S. alba* and *L. speciosa* both underwent the WGT event, they are currently diploid. Chromosome evolution analysis reveals that the mangrove species has a reduced number of chromosomes and undergoes more chromosome rearrangements compared to *L. speciosa*. In fact, polyploids may face substantial disadvantages, including redundant components, gene dosage imbalance, increased replication and metabolic costs, cellular mismanagement, and a higher propensity for polyploid mitosis and meiosis to produce aneuploid cells. Despite these immediate challenges, some polyploid lineages have persevered and even thrived. We have discussed the advantages and disadvantages of polyploidy and diploidy, as well as the polyploidization–rediploidization process in the Section (Line 218-238).

8. Line 204-205. Please add the references to this point (“all angiosperms have undergone successive rounds of polyploidization and rediploidization process”).

[Reply and Revision]: Thanks for your suggestion. We added the references in the revision.

References:

4. Van de Peer, Y., Mizrachi, E. & Marchal, K. The evolutionary significance of polyploidy. *Nat Rev Genet* **18**, 411–424 (2017).
10. Jiao, Y. et al. Ancestral polyploidy in seed plants and angiosperms. *Nature* **473**, 97–100 (2011).
11. Jiao, Y. et al. A genome triplication associated with early diversification of the core eudicots. *Genome Biol* **13**, R3 (2012).
83. Myburg, A. A. et al. The genome of *Eucalyptus grandis*. *Nature* **510**, 356–362 (2014).

9. Line 257-296. By conducting transcriptome analysis, the authors identified WGT-retained genes related to adaptive evolution. In addition to the presentation of figures and tables, please provide a concise summary of these identified genes, along with their sequences. It could facilitate the utilization.

[Reply]: Thanks. We have included detailed information and insights about these genes in Supplementary Table 11 and Fig. 5. And we have uploaded their sequences to NGDC database under BioProject ID PRJCA005319. Please refer to Line 559-564 for further details.

10. Line 400-401. What model was used in the MCMC analysis?

[Reply]: Thank you for the comments. We did not explain the model clearly. The revision below is much more explicit and accurate about the MCMC analysis.

[Revision]: (Line 434-437) Following its reconstruction, we estimated the divergence time among the

seven species using MCMCTREE from the PAML (v4.9j) package with approximate likelihood calculation^{134,135}. The HKY85+G nucleotide substitution model and independent-rates clock model were employed in the molecular dating.

11. Line 505. The abbreviation of “confidence intervals” should be “CIs”.

[Reply]: We appreciate your attention to detail. We have updated the abbreviation for “confidence intervals” to “CIs” in the revision.

12. Fig. 2d looks like a partially collinear relationship. Kindly make a note of this observation in the figure legend.

[Reply and Revision]: Thanks. We have added the following sentence to the legend (Line 948-949): “The representation showcases partial regions of the genomes.”

13. Line 880. “Supplementary Table S7” should be “Supplementary Table 7”.

[Reply]: We have corrected it.

14. Please add a row of anchored gene numbers to Table 1.

[Reply]: We have added a row of anchored gene numbers to Table 1.

15. Please state the source of the pictures (a, b, c) in Supplementary Figure 3.

[Reply]: The pictures (a, b, c) in Supplementary Figure 3 were taken by the author. A corresponding note has been included in the legend of Supplementary Figure 3.

16. Please change “Hi-C” to “Hi-C reads” in Supplementary Table 1.

[Reply]: We thank the reviewer for carefully handling our manuscript. We have corrected it and checked the manuscript carefully.

Reviewer #2 (Remarks to the Author):

This is a very interesting manuscript taking a novel paired rediploidisation system. It presents a unique look at the rediploidisation process over medium-scale (not recent) evolutionary time. It will be of significance to workers in several fields, from ecology and evolution to fundamental plant biology. In general Does the work supports the conclusions and claims and I have no major comments that require modification of the analysis; as far as I can ascertain, the methodology is sound. I provide more minor comments below with the intention of helping to enhance the presentation.

[Reply]: Thanks for the positive comments.

26: ‘recent’: 64 million years ago is not what most in the field would call a recent polyploidy event. Nor is it necessary for the novelty here to suggest this is recent, so I propose to strike all use of ‘recent’.

[Reply]: Thanks for your suggestion. We have removed the “recent” used to modify whole genome triplication (WGT) from both the main text and supplementary information.

32: ‘adaptive value’ can be inferred by retention, but ‘accelerated evolutionary adaptation’ cannot.

[Reply]: Thanks for your useful suggestion. We have revised related sentences in the revision.

[Revision]: (Line 27-29) Additionally, we observe strong selection acting on three-copy retentions following the polyploidization–rediploidization process, indicating the potential adaptive value in response to new environments.

(Line 282-285) Collectively, these results indicate preferential retentions of three-copy genes following the polyploidization–rediploidization process, driven by strong selection and possessing potential adaptive value in response to new environments.

34: ‘we propose a model’: if you say you propose a model in the abstract, say something about it other than selling the quality of your genomes that served as input data. I also note that the model in the figure is a little bit ‘light’; the novelty of the ‘model’ is to me not great, so I would suggest stressing the new model much, to be frank.

[Reply and Revision]: In this study, genomic evidence supporting polyploidization and rediploidization is provided through chromosome-scale comparative genomic analyses. With these insights, we can improve a model that elucidates the process of polyploidization–rediploidization in plants during global climate change, addressing both macro (a) and micro (b) perspectives. We have now incorporated genomic evidence for rediploidization, depicting changes at both the chromosome

and gene levels in Fig. 3. This revision better reflects the central theme of our paper. Refer to Fig. 3 and Line 232-238 for detailed information.

54-57: this gives the false impression that WGD is an adaptive panacea when in fact most young polyploids are not stable. This downside can be clearly seen in discussion in e.g. <https://doi.org/10.3389/fevo.2018.00117>

[Reply]: We appreciate the reviewer's suggestion. We have revised this part to address the potential misconception. The updated content now includes the disadvantages of WGD and then presents some cases about the adaptive potential of polyploidy, especially in the context of dynamic and fluctuating environmental conditions. The revised text is pasted below.

[Revision]: (Line 47-53) Despite the challenges that have emerged since the WGD, such as stable chromosome segregation, detrimental ecological interactions with diploid progenitors, and minority cytotype exclusion^{35,36}, the polyploidy events observed in plants highlight their evolutionary potential. Experimental and simulation studies have supported the adaptive potential of polyploidy, especially in the face of dramatic and fluctuating environmental conditions³⁷⁻³⁹. Overall, polyploidy has been recognized as a major driving force behind evolutionary adaptation and diversification^{4,5}.

58: 'Plants on the Earth': suggest rephrase to 'Plants' or some other formulation; it's awkward as is.

[Reply and Revision]: We have replaced "Plants on the Earth" with "Plants" in the revision. Please see Line 54.

66: 'rain forests of the sea' is puzzling to me and seems not relevant. Perhaps it's a commonly used term for these species, but seems to me a little bit misleading. Rainforests are much more species-diverse in my mind than mangroves.

[Reply]: We appreciate the reviewer's perspective, and we have removed the phrase "known as the 'rainforests of the sea'" as suggested. Please see Line 62.

78: references 51 and 52 are introduced as examples or support for the statement that high quality genomes have provided opportunity to study genomes post WGD. Immediately following this statement the current study is introduced, but there is no suggestion what the gap was following studies 51 and 52 or what the major current innovation motivating this study in relation to them. It would be good to say here how the current study contributes something truly novel.

[Reply]: WGD events have played a significant role in the evolutionary history of angiosperms, and it has been proposed that these events contribute to plant adaptation and survival in the face of environmental changes. However, empirical evidence for rediploidization has been lacking, particularly at the genomic scale. By reconstructing ancestral genomes and inferring the trajectory of plant genome evolution, our study innovatively explores the rediploidization process following polyploidization on a genomic scale. In the revision, we have added a sentence to connect them. We appreciate the reviewer's suggestion.

[Revision]: (Line 71-74) With advancements in genome sequencing and assembly technologies, high-quality chromosome-scale genomes provided an opportunity to reconstruct ancestral genomes and infer the trajectory of plant genome evolution^{60,61}. We can now explore the process of rediploidization following polyploidization on a genomic scale. In this study, ...

83-84: 'The genome sequences... evolutionary research in plants' is far too weak a statement to make any interest. Suggest rewriting the end of the intro to be more powerful, stating the specific point of this study, as it currently is very weak indeed.

[Reply]: In the revision, we have rephrased the end of the introduction to highlight the central theme of the manuscript.

[Revision]: (Line 77-79) Through comprehensive analyses, we trace the evolutionary history of genomes and investigate the polyploidization–rediploidization process and its implications for adaptive evolution in the face of global climate change.

88: it is strange to start here with HiC. What about the sequencing of the contig assembly? There is no reference given for this, so where do I go to learn about how it was originally sequenced?

[Reply]: Thank you for the comments. We did not explain the assembly clearly. To address this, we have included additional information and a reference regarding the contig assembly of *S. alba*.

[Revision]: (Line 83-86) We first utilized high-throughput chromosome conformation capture (Hi-C) technology to improve the genome of *S. alba*. This improvement builds upon our prior study utilizing PacBio Single-Molecule Real-Time (SMRT) sequencing and Illumina short reads sequencing⁵⁴, resulting in a chromosome-scale assembly (Supplementary Table 2).

Reference

54. He, Z. *et al.* Convergent adaptation of the genomes of woody plants at the land–sea interface. *Natl Sci Rev* 7, 978–993 (2020).

90: '204.46 Mb': what was the expected size based on flow or kmers? I see these data are the methods; it would be better if they were in the results here too/instead.

[Reply]: In our previous study, we assembled the genome of *S. alba* using PacBio SMRT sequencing and Illumina short-read sequencing. The assembled genome size is 207.20 Mb, consistent with the genome size estimated by k-mer-based analysis (211.67 Mb). After eliminating a small number of redundant sequences and anchoring to chromosome scale, the newly assembled genome derived from anchored contigs measured 204.46 Mb.

[Revision]: (Line 86-89) The newly assembled genome derived from anchored contigs was 204.46 Mb, aligning closely with the genome size estimated through k-mer-based analysis (211.67 Mb). It comprised 12 chromosomes (97.60% of all sequences) and 40 unanchored scaffolds.

96: 'consistent with the estimated genome size': what is that estimate?

[Reply]: We have incorporated details regarding the genome size estimations of *L. speciosa* through both flow cytometry and k-mer-based analysis.

[Revision]: (Line 92-94) The assembled genome of *L. speciosa* was 319.66 Mb, with an N50 value reaching 12.74 Mb, consistent with the estimated genome size (361 Mb by flow cytometry and 340.46 Mb by k-mer-based analysis).

100: a sentence about how the annotation was done would be nice in the results.

[Reply]: Thank. We have added sentences about the annotation in the revision.

[Revision]: (Line 97-102) The gene prediction process involved a comprehensive approach, combining *ab initio*, homology-based and RNA-seq-assisted strategies. The integration of these predictions through EvidenceModeler resulted in the identification of non-redundant and consensus gene models for the *S. alba* and *L. speciosa* genomes (see Methods for details). This unveiled a total of 25,284 (Supplementary Fig. 5) and 30,497 (Supplementary Fig. 6) protein-coding genes, respectively, characterized by high completeness (Supplementary Table 3).

107: suggest to rephrase to either 'Fewer TEs accumulate' or 'Less TE accumulation'

[Reply]: We have revised the text based on your recommendation.

[Revision]: (Line 108) Less TE accumulation in the mangrove

123: ‘taxa’ is plural; ‘taxon’ is singular, which is what I think you want here.

[Reply and Revision]: Thanks for your comment. We have utilized “taxon” instead of “taxa” in the revision. Please see Line 125.

144, 5, 7, 152, 171: ‘recent’: this happened at around the time of the K-Pg boundry! I would disagree with the use of the word ‘recent’, and besides it does no benefit to call this WGT ‘recent’ for your story. You don’t need it.

[Reply]: Thanks for your suggestion. We have removed the “recent” used to modify whole genome triplication (WGT) from both the main text and supplementary information.

197-8: There is evidence of increased signal of WGD at times of dramatic environmental and climate change, but there is absolutely not sufficient evidence to make such a strong statement that ‘newly formed polyploids typically possess a significant fitness advantage over diploids.’ Typically neopolyploids are dead due e.g. in autotetraploids to meiotic catastrophe. I am passionate about polyploids and have devoted my career to studying them and I therefore also want to say this, but this statement goes much farther than the data and must be revised. See a more balanced discussion of this in e.g. the review I reference in 54-57 above. One much more appropriate suggestion would be to say: “newly formed polyploids can possess fitness advantages over diploids” or more accurately something like “signals of WGD persistence correlate with times of environmental and climate change, suggesting potential benefit to WGD in the face of challenges”.

[Reply]: We agree with the reviewer’s viewpoint. We have revised this sentence by the suggestion.

[Revision]: (Line 219-222) During periods of dramatic global environment and climate change, newly formed polyploids can possess fitness advantages over diploids. This is supported by evidence that the persistence of WGD correlates with times of environmental and climate change, suggesting potential benefit for the WGD in the face of challenges^{4,35,45,74–77}.

201: ‘will’: here too the language is too simplistic. I would suggest ‘may’ and to again see the review I reference above for more balanced examples.

[Reply]: We have revised this sentence based on your recommendation.

[Revision]: (Line 226-228) As climatic conditions stabilize and environmental conditions improve, polyploids may experience reduced fitness compared to diploids due to the accumulation of genetic load, increased mutational load, slower positive selection, and reduced growth rates^{35,37,81,82}.

197-202: the studies referenced here are not strong and often rather old. Again, the statements should be more moderate and less absolute and referencing various excellent reviews of this broad literature in addition to the one I suggest would be much more informative: e.g. doi.org/10.1086/700636 and [doi: 10.1101/gad.271072.115](https://doi.org/10.1101/gad.271072.115).

[Reply]: Thanks for your suggestion. Incorporating the literature you kindly provided, we have revised the text to provide a more comprehensive and balanced perspective on the advantages and disadvantages of polyploidy and diploidy.

[Revision]: (Line 219-228) During periods of dramatic global environment and climate change, newly formed polyploids can possess fitness advantages over diploids. This is supported by evidence that the persistence of WGD correlates with times of environmental and climate change, suggesting potential benefit for the WGD in the face of challenges^{4,35,45,74-77}. Nevertheless, polyploids may also face substantial disadvantages, including redundant components, gene dosage imbalance, increased replication and metabolic costs, cellular mismanagement, and a higher propensity for polyploid mitosis and meiosis to produce aneuploid cells^{35,58,77,78}. Despite these immediate challenges, some polyploid lineages have persisted and even thrived^{79,80}. As climatic conditions stabilize and environmental conditions improve, polyploids may experience reduced fitness compared to diploids due to the accumulation of genetic load, increased mutational load, slower positive selection, and reduced growth rates^{35,37,81,82}.

265: 'functional analysis': do you mean GO analysis? This is unclear in the text. Describing this analysis better in the main text is needed. The results seem interesting, but they need to be better shown. Nor do I see a clear description of this analysis in the methods.

[Reply]: The functional analyses encompass GO enrichment and gene function assessments. We have revised the sentence for clarity. For gene function assessments, we summarized key pathways from the literature and matched genes based on annotations. For GO enrichment analysis, we identified single-copy genes using the `duplicate_gene_classifier` module from MCScanX. We performed GO enrichment analysis of two-copy and three-copy retention groups after the WGT event with single genes as a control using BiNGO in Cytoscape (v.3.7.2). It is available in Supplementary Note.

Why is GO analysis in the Supplementary Information? In this section, our focus is on uncovering the mechanisms behind the adaptive traits of the mangrove tree, specifically waterlogging and salt tolerance. While GO enrichment analysis provides general insights, we place greater emphasis on WGT-retained genes and the pathways associated with adaptive traits. We identify specific genes and discuss them in detail within this section. The pathways of gene retention after the WGT events are more interesting and relevant, offering clearer insights than GO enrichment. This aspect is also more

crucial. Therefore, in this section, we elaborate on specific pathways and genes. Conversely, GO analysis is presented in the Supplementary Information. I hope you can understand my intention.

[Revision]: (Line 295-298) Therefore, we conducted functional analyses among the retained genes, which encompassed GO enrichment (Supplementary Fig. 25, Supplementary Note) and gene function assessments based on annotations. Our focus was particularly directed toward the 584 three-copy retention groups generated by the WGT event.

Between the standard GO analysis and the ‘functional analysis’ not well described, it seems the enriched terms are rather ‘high level’ and not very descriptive: e.g. ‘regulation of biosynthetic process’, ‘regulation of metabolic processes’, ‘biological regulation’. Can the authors better highlight the more specific GOs that are better describing particular processes or functions?

[Reply]: As explained in the previous response, our GO enrichment analysis may not distinctly elucidate results pertaining to adaptation. To address this limitation, we integrated transcriptomes from salt gradient experimental treatments in *S. alba*. This enabled the identification of expression patterns for key WGT retained genes across salinity conditions, shedding light on the mechanism underlying salt tolerance.

268-9: these are of course very interesting pathways!

[Reply]: Thank you for your positive feedback and encouragement. We have included detailed information and insights about these genes in Supplementary Table 11 and Fig. 5. Additionally, we have uploaded their sequences.

298: no need to say ‘highly accurate’

[Reply]: Thanks. We have removed it in the revision.

The conclusion is much better without the last sentence to be honest. There is no need at all for such a broad statement, which reduces the impact of the rest of the nice writing.

[Reply]: Thank you for your suggestion. We have rewritten the last sentence in the conclusion section of the revised manuscript.

[Revision]: (Line 338-339) Overall, our study contributes valuable insights into the plant evolution.

319: more details should be given on the CTAB and reference it.

[Reply]: Thanks. We used the CTAB method for the DNA extraction. We have incorporated a relevant reference, and the revision is much more explicit and accurate.

[Revision]: (Line 350-351) High-molecular-weight (HMW) genomic DNA was isolated from *L. speciosa* leaf tissue using the CTAB (hexadecyltrimethylammonium bromide) method¹⁰⁷ for both PacBio Single-Molecule Real-Time (SMRT) long-read sequencing and Illumina short-read sequencing.
107. Doyle, J. J. & Doyle, J. L. A rapid DNA isolation procedure for small quantities of fresh leaf tissue. *Phytochem Bull* **19**, 11–15 (1987).

Reviewer #3 (Remarks to the Author):

The authors sequenced and assembled genomes of two related species from the family Lythraceae. One species belongs to mangrove species, while the second species (*Lagerstroemia speciosa*) is not adapted to the mangrove ecosystem. Overall the paper is clearly structured and well-written, however I need to review this work in comparison with similar studies published in Nat Comm and similar journals recently. In this comparison I found the present paper to contain less novelty than some other genome-based papers published in Nat Comm. Here are some key points I considered as critical:

[Reply]: We appreciate the helpful comment.

(1) Page 3. „almost all mangrove species are currently considered diploids“. Please make sure that the current interpretation is correct. I did not check the cited references, however very often species are considered diploid due to the lack of other (lower) chromosome numbers in a given taxon (e.g., genus), and thus, species of *Helianthus* ($n=17$) are or were considered to be diploid (and there are more examples of course).

[Reply and Revision]: Thanks. We understand the reviewer's concern. We have conducted a literature review and summarized karyotype and ploidy information for 23 typical mangrove species, placed in 11 families, to substantiate the interpretation. We have incorporated the data into Supplementary Table 1 in the revision.

(2) Page 4, line 109... Here the authors are reporting the difference in the proportion of TEs between the two genomes, not really appreciating the fact that logically 12 centromeres will have much less repeats than 24 centromeres (*Lagerstroemia*). And so it is also logic to expect that the mangrove species will have smaller nuclear genome than the inland species, as it was found. This part is overall trivial and concluding that plants and the investigating species have LTR retrotransposons. Where are other types of repeats, including tandem repeats? The concluding sentence is somewhat naive but logic – see above – the authors should, for example, consider that logically there are less insertions of LTR-RTs in the mangrove genome due to the lower number of genomic regions where the insertions can be tolerated (typically these are pericentromere regions: 12 versus 24).

[Reply]: The number of chromosomes and centromeres may influence the size of transposable elements (TEs) but has little effect on the proportion of TEs. Upon comparison, we found that *S. alba* not only possesses fewer TE sequences (43Mb vs 117Mb) but also has a smaller proportion of TEs (20.95% vs 36.50%). Long terminal repeat retrotransposons (LTR-RTs), often exhibit a higher copy number and larger size in plant genomes, significantly contributing to genome size growth⁶². On one hand, LTR-RTs have a larger size in plant genomes compared to Short Interspersed Nuclear Elements

(SINEs), Long Interspersed Nuclear Elements (LINEs), DNA elements, and other repeats. On the other hand, intact LTR-RTs can be used to estimate the distribution of insertion time providing insights into the evolutionary history. Therefore, in this context, we compare genome size, accumulation of TEs, and rate of LTR-RT insertion to elucidate the more simplified genome of *S. alba*. Additionally, we have included information about the fewer chromosomes in *S. alba* in the revised version as suggested by the reviewer.

[Revision]: (Line 111-112) First, we observed that *S. alba* has fewer chromosomes compared to *L. speciosa* (Fig. 1b).

(Line 119-121) Overall, the mangrove species *S. alba* maintains a smaller genome size, fewer chromosomes, lower accumulation of TEs, and a reduced rate of LTR-RT insertion, resulting in a more simplified genome.

(3) Page 5. The phylogeny. With so many genome sequenced and analyzed phylogenetically I cannot be really happy about the phylogeny including 7 species (out of 250 000 as the authors correctly report in the Introduction)! It is also unclear and not discussed if their divergence time estimates are congruent with already published datings.

[Reply]: Thanks. Before we answer the reviewer's instructions, we should say that the main results in this study are based on two chromosome-scale genomes from Lythraceae plants.

The phylogenetic analysis involving seven species with the availability of chromosome-scale reference genomes allows for the accurate estimation of the divergence time between *S. alba* and *L. speciosa*, which is a basis for determining the timing of the shared WGT event. Following the reviewer's suggestion, we assess the congruence of the divergence times estimated in this study with those reported in previous studies, as summarized in the TimeTree database. The comparison indicates a general convergence of results. Furthermore, we expanded our phylogenetic analysis by constructing a larger-scale tree, including 42 sequenced angiosperms from 39 orders, along with the gymnosperm *Gnetum montanum* as an outgroup. The revised text is provided below.

[Revision]: (Line 131-137) The divergence times were consistent with previous studies (Supplementary Table 5)⁶⁷. Additionally, our estimation suggests that the mangrove *S. alba* diverged from the closely related inland woody plant *L. speciosa* around 57.79 Mya, while the common ancestor of them diverged from the same family plant *P. granatum* around 67.82 Mya (Fig. 2a). We further constructed a larger-scale phylogenetic tree, incorporating 42 sequenced angiosperms along with the gymnosperm *Gnetum montanum* (as an outgroup), to reflect the positions of these plants within Lythraceae (Supplementary Fig. 10).

(Line 440-449) In order to delineate the positions of these plants within Lythraceae, we expanded our

analysis by constructing a more extensive phylogenetic tree using these seven plants, other 35 genome-sequenced angiosperms, and the gymnosperm *Gnetum montanum* as an outgroup (Supplementary Table 14). Utilizing the embryophyta_odb10 lineage ancestral variant dataset (comprising a consensus sequence and variants of extant sequences) in BUSCOv5¹²⁵, we identified 868 low-copy nuclear genes. We then performed sequence alignment and phylogenetic inference as described earlier. The early divergence times in angiosperms were set to 125–247.2 Mya^{138,139}. All MCMC analyses were independently run twice to ensure convergence, with 10 million generations and sampling every 500 generations after a burn-in of 1,000,000 iterations. The phylogenetic trees were visualized using the R package GGTREE¹⁴⁰.

(4) The same page, line 144. „219 syntenic block pairs comprising 3,333 gene pairs in *P. granatum*“. And later: „suggesting that *P. granatum* did not experience recent polyploidy events.“ It remains to be explain where from the 219 syntenic block pairs in *P. granatum* come from.

[Reply]: Due to variations in the number of genes within syntenic blocks, gene pairs can provide a more robust response to polyploidy events. Comparatively, *P. granatum* exhibits fewer gene pairs than *S. alba* and *L. speciosa*. The Ks distribution reveals peaks in *S. alba* and *L. speciosa*, but not in *P. granatum*, indicating that *P. granatum* did not experience the polyploidy events. The syntenic block pairs and gene pairs in *P. granatum* originate from segmental duplication and paleopolyploidy events, such as the γ -WGT event associated with the early diversification of core eudicots.

(5) The usage of „recent“ for the identified whole-genome triplication 64 mya is really funny and I did not understand why the authors actually use „recent“ for this very old polyploidization event.

[Reply]: Thanks for your suggestion. We have removed the “recent” used to modify whole genome triplication (WGT) from both the main text and supplementary information.

(6) Page 6. Line 156 and following. I really did not understand where from the authors get information how the WGT helped to cope with the K-Pg global extinction and later events? If I accept this tale, then it comes even before adaptive genes are introduced, researched and discussed.

[Reply]: Thanks. There is previous research on how the WGT helped to cope with the K-Pg global extinction. We have integrated relevant references and discussed this aspect in the revision. The revised text is provided below.

Regarding the role of WGT in dealing with subsequent events, we elaborate on this part in the “WGT retained duplicates for root development and salt tolerance” section. We present genomic and

transcriptomic evidence supporting the notion that WGT-retained duplicates contribute to adaptation to intertidal zones.

[Revision]: (Line 166-172) Polyploidy events play a significant role in reshaping gene regulatory networks in response to environmental stresses^{9,71}. A series of ancient WGD events occurred independently in numerous plant lineages around the K-Pg boundary^{43,45,49}. These events served as a buffer for plants, enhancing their ability to survive and adapt to rapidly changing environments by increasing genomic plasticity and generating diverse genotypic combinations. The WGT event, in particular, likely contributed to the survival of plants during the extinction event.

(7) The same page: „If we consider only polyploidy, the haploid chromosome number would be 168 (21*2*2*2).“ It is not clear whether this is related to *Utricularia* or Lythraceae and, second, it has to be explained where 21 comes from (I guess it could be 3 x 7 of the Gamma WGT?).

[Reply]: Thanks. We understand the reviewer's concern. *Utricularia gibba* indeed has a small plant genome, as documented in the *Nature* (2013) publication. We introduced this information to provide an intuitive understanding of the impact of polyploidy on chromosome number. This paves the way for the chromosomal evolution of our research plant below. We have included specific species information and referenced the ancestral chromosome number ($n = 7$). The revised text is presented below.

[Revision]: (Line 189-193) For example, *Utricularia gibba*, despite having a small plant genome, has a haploid chromosome number (n) of 14, yet it has undergone three WGD events since the well-known γ event shared by core eudicots¹⁶. If we exclusively consider polyploidy, the haploid chromosome number of *Utricularia gibba* would be $7*3*2*2*2$ or $n = 168$, based on the ancestral chromosome number ($n = 7$) before experiencing γ -WGT event⁷³.

(8) Page 7, 1st paragraph. To me, the fact that the ancestral genome had 8 chromosome pairs is repeated three times. This part is very vague and general – the reader can look at figure 2f but what can be seen and inferred from this karyograms? It is also not very clear that actually the inland genome has retained the ancestral number of chromosomes after the WGT, but there are chromosomal rearrangements (what types of chr. rearrangements?). Suppl fig 13: why *Pemphis acidula* with the duplicated ancestral genome is not mentioned and used for comparisons with *P. granatum* and the two species investigated in here?

[Reply]: The previous analysis based on conserved collinear genes was fragmented and lacked clarity. To enhance data presentation, in the revision, we utilized a new software, WGDI (a user-friendly toolkit for evolutionary analyses of whole-genome duplications and ancestral karyotypes) to identify

adjacent conserved collinear blocks among all chromosome pairs and reconstruct the Ancestral Lythraceae Karyotype (ALK). In addition to characterizing the state of ancestral chromosomes, we also inferred the chromosome histories of both species, particularly illustrating a probable karyotype evolution that captures the complexity of the evolutionary history of chromosomes in *S. alba* (Supplementary Fig. 18).

We also conducted synteny analysis among the modern genomes of the three Lythraceae species and identified numerous chromosome rearrangements. In the genome of related inland species, analyses of chromosome evolutionary history and synteny in the modern genomes show that intra-chromosomal inversions predominantly govern chromosome changes (Fig. 2f and Supplementary Fig. 19).

This article only utilized the chromosome number information of *Pemphis acidula*, solely for inferring the chromosome number of the ancestral ALK. Compared to *Pemphis acidula*, *P. granatum* possesses a high-quality genome and a relatively simple evolutionary history (lacking recent polyploidization), making it a more suitable outgroup for this study.

[Revision]: (Line 199-216) Then, we reconstructed the ancestral Lythraceae karyotype (ALK) using WGDI based on adjacent conserved collinear blocks. Our evolutionary scenario suggests that the ALK of *S. alba*, *L. speciosa*, and *P. granatum* genomes consisted of eight proto-chromosomes with 18,885 proto-genes. As shown in Fig. 2e, the ancestor underwent a WGT event and subsequently experienced chromosomal rearrangements to attain their modern genome structure. The chromosome origin of *S. alba* appears more intricate than that of *L. speciosa*. *S. alba*'s chromosomes underwent a greater number of fission and fusion events compared to *L. speciosa*, although intra-chromosomal inversions were common in the chromosome histories of both species (Fig. 2e and Supplementary Fig. 17). Due to the complexity of chromosome evolutionary history in *S. alba*, we illustrated it using reciprocally translocated chromosome arms (RTA), end-to-end joining (EEJ), nested chromosome fusion (NCF) events, fission events, and chromosome inversions to depict a probable karyotype evolution (Supplementary Fig. 18).

Although the reconstructed ancestral karyotype is highly likely to possess a structure very similar to the true ancestral genome, it may not be entirely identical⁶⁰. Furthermore, we performed synteny analysis among the modern genomes of the three Lythraceae species and confirmed numerous chromosome rearrangements (Fig. 2f). In contrast to intra-chromosomal inversions observed in related inland species, *S. alba* exhibited significant fission and fusion events (Supplementary Fig. 19). These findings indicate that the mangrove species has a reduced number of chromosomes and undergoes more chromosomes rearrangements compared to its closely related inland species *L. speciosa*.

(Line 516-525) We utilized WGDI (v0.6.5) to identify adjacent conserved collinear genes and blocks among all chromosome pairs within the three Lythraceae species, and then reconstructed the Ancestral

Lythraceae Karyotype (ALK), excluding interference from fragmented collinear regions, following the tutorial^{152,153}. Subsequently, we visualized the global pattern of chromosomal changes in extant species. Furthermore, we depicted the evolutionary history of *S. alba* chromosomes to provide a clearer representation of the karyotype evolution¹⁵². While the reconstructed ancestral karyotype almost certainly has a very similar structure to the true ancestral genome, it may not be absolutely identical⁶⁰. We also conducted synteny analysis among the modern genomes of the three Lythraceae species using MCScanX and JCVI to discover chromosome rearrangements¹⁵⁴.

(9) I missed more indepth analysis how the hexaploid genome was rearranged in the two species investigated.

[Reply]: Thank you for the comment. In response to the reviewer's request, we conducted a comprehensive re-analysis of karyotype evolution as well as a synteny analysis among the modern genomes, to depict and compare the patterns of chromosome rearrangement in both the mangrove species and the related inland species. In karyotype evolution analysis, distinguishing different copies of ancestral chromosomes resulting from polyploidization is difficult and controversial. To address the complexity associated with the issue, we opted for a more straightforward approach. Instead of focusing on the hexaploid genome ($n = 24$), we utilized the completeness of the ALK ($n = 8$). This approach not only simplifies the presentation, enhancing clarity, but also guarantees the presence of all non-homologous chromosomes, ensuring accuracy in our depiction of the karyotype evolutionary history. The revisions pertaining to this aspect have been integrated into the manuscript, providing a more detailed and explicit account of the chromosomal rearrangement patterns. These changes can also be found in the revision of the previous question.

(10) The polyploidization-diploidization cycle (Figure 3) was published by several authors (some cited, some not) and, forgive me, to call this simple and already several times published scheme as A MODEL is little too much. The authors should eliminated „this model“ or modify it by acknowledging researchers who published basically the same cycle shemes earlier.

[Reply and Revision]: This model elucidates the polyploidization–rediploidization process in plants during global climate change, addressing both macro (a) and micro (b) perspectives. The robust support for our hypothesis is exemplified by the case of mangrove species. We have now incorporated genomic evidence for rediploidization, including changes at both the chromosome and gene levels in Fig. 3. Nevertheless, we sincerely acknowledge your suggestion to use “improve the model” rather than “propose a new model”. We also cite relevant findings and viewpoints from previous studies. Please see Fig. 3 and Line 232-238 for details.

(11) Expression analysis. I was really not sure how this was meant and what exactly can be concluded when the same analysis was not done for the inland species and/or another (non-)mangrove species for which comparable dataset is available.

[Reply]: Expression analysis was conducted to identify the differential expression of paralogous gene pairs generated by the WGT event. Our findings reveal that approximately 60% of these paralogous gene pairs were differentially expressed across the four tissues in the mangrove species, suggesting potential neo- and sub-functionalization of the retention genes following the polyploidization–rediploidization process. In the revision, we extended our expression analysis to newly sequenced RNA-seq data from various tissues of the closely related inland plant *L. speciosa*. Employing the HISAT2–HTSeq–exact conditional test workflow, we identified a similar pattern, with around 60% of the paralogous gene pairs resulting from the WGT exhibiting differential expression across the tissues in the related species, mirroring our observations in the mangrove species. These comprehensive data and results provide further support for the potential neo- and sub-functionalization of the retention genes following the polyploidization–rediploidization process.

[Revision]: (Line 259-264) Similarly, we explored the expression divergence of WGT retained genes in the closely related inland plant *L. speciosa*. We also identified that around 60% of the paralogous gene pairs resulting from the WGT exhibited differential expression across four tissues in the related species (Supplementary Tables 9-10), mirroring findings in the mangrove species. These results suggest the potential neo- and sub-functionalization of the retention genes following the polyploidization–rediploidization process.

(Line 536-539) Additionally, we performed RNA-seq on leaf, stem, flower, and fruit tissues of *L. speciosa* (Supplementary Table 9 and Supplementary Fig. 29). We employed the HISAT2–HTSeq–exact conditional test workflow, as described earlier, to identify differentially expressed duplicated gene pairs.

(12) Also, surprisingly, I did not get any information how the hexaploid genome has been formed and if post-polyploid gene fractionation (diploidization process in general) impacted all three subgenomes in the same or different way.

[Reply]: The hexaploid genome is formed through a process known as whole-genome triplication (WGT). This event originates from hybridization between tetraploid and diploid species¹⁷⁻¹⁹, as outlined in the Introduction section (Line 44-45). To further illustrate this process, a schematic diagram has been included in Fig. 3b.

The WGT event results in the generation of three subgenomes. However, distinguishing between these subgenomes without the presence of ancestral species of tetraploid and diploid remains an

unsolved challenge, especially when the WGT event is distant. Subgenome analysis is typically performed in allopolyploid plants such as *Brassica* (Cheng *et al.*, 2016), *Gossypium* (Chen *et al.*, 2020), and *Nicotiana* (Ranawaka *et al.*, 2023), where ancestral species information is available. In our study, we explore the rediploidization process of the whole genome, utilizing genomic evidence at both the gene level (examining sequence divergence and expression divergence) and the chromosome level (assessing chromosomal rearrangements, including fission, fusion, and inversions). Using a mangrove species as an example, we demonstrate how the rediploidization process contributes to its adaptation to the intertidal environment.

References

- Cheng, F. *et al.* Subgenome parallel selection is associated with morphotype diversification and convergent crop domestication in *Brassica rapa* and *Brassica oleracea*. *Nat Genet* **48**, 1218–1224 (2016).
- Chen, Z. J. *et al.* Genomic diversifications of five *Gossypium* allopolyploid species and their impact on cotton improvement. *Nat Genet* **52**, 525–533 (2020).
- Ranawaka, B. *et al.* A multi-omic *Nicotiana benthamiana* resource for fundamental research and biotechnology. *Nat Plants* **9**, 1558–1571 (2023).

We appreciate all the comments and suggestions.

Reviewers' Comments:

Reviewer #1:

Remarks to the Author:

I am fully satisfied with their response and revisions. The manuscript can be accepted

Reviewer #2:

Remarks to the Author:

The authors have done an excellent job with this very interesting study.

I am fully satisfied that my suggestions have been appropriately incorporated into this improved draft. I only provide below several minor textual suggestions.

Line 151 strike 'these groups'

Line 171 suggest slightly softer language such as: "we suggest that the WGT events may have contributed to the survival"...

Line 182 suggest change 'remarkable' to 'powerful' or similar

Line 199 suggest some more clear topic sentence to introduce the paragraph rather than 'Then, we...'

Line 215 suggest 'undergoes' to 'underwent'

Line 219 suggest strike 'global'; it need not be 'global'

Line 229 suggest change 'is crucial for' to 'may be inevitable' to make it a little bit less sure.

Line 264 suggest 'retention' to 'retained'

Line 268 suggest change 'to shape into' to 'to aid'

Reviewer #3:

Remarks to the Author:

The authors reflected on most comments of all referees.

I refrain from commenting small issues, which could be still debated, modified and improved. My major concern is still the surprising lack of comments on the presumable mode of the origin of the hexaploid ancestral genome. I understand what the authors state in their rebuttal, namely that the WGT is old (65 mya) and that it is not easy to dissect the three subgenomes. Well, this is probably the case in the reshuffled genome of the mangrove species ($n=12$ chromosomes), but is this also the case in Lagerstroemia species with triplicates of all 8 chromosomes ($n=24$) more or less still preserved (Fig. 2e)? I still think that the authors should provide evidence for the failure of subgenome phasing (i.e. explain why this is impossible) or do this type of (standard) analysis. This is done, for example, based on gene densities in triplicated chromosome regions. Note that "gene fractionation" is mentioned in figure 3b, but I was not able to find any mention of biased or unbiased gene fractionation in the text. In fig. 3b, I did not understand what is "Fractionation, Sequence, Expression". Precisely, I did not understand to what terms "sequence" and "expression" refer. Also, the usage of "n" and "x" is incorrect in this figure. Note that each eukaryotic organism produces haploid gametes (with n chromosomes), diploid soma has $2n$ chromosomes. However ploidy levels equal to x -folds (diploid $2x$, triploid $3x$, tetraploid $4x$). Note that following the logic in the fig 2b, the outcome of diploidization is $4n$ (not $2n$).

Replies to reviewers' comments point by point:

Reviewer #1 (Remarks to the Author):

I am fully satisfied with their response and revisions. The manuscript can be accepted.

[Reply]: Thank you for your positive feedback and for taking the time to review our manuscript. Your feedback has been invaluable in improving the quality of our work. We are delighted to hear that you are fully satisfied with our response and revisions.

Reviewer #2 (Remarks to the Author):

The authors have done an excellent job with this very interesting study.

I am fully satisfied that my suggestions have been appropriately incorporated into this improved draft.

I only provide below several minor textual suggestions.

Line 151 strike 'these groups'

Line 171 suggest slightly softer language such as: "we suggest that the WGT events may have contributed to the survival"...

Line 182 suggest change 'remarkable' to 'powerful' or similar

Line 199 suggest some more clear topic sentence to introduce the paragraph rather than 'Then, we...'

Line 215 suggest 'undergoes' to 'underwent'

Line 219 suggest strike 'global'; it need not be 'global'

Line 229 suggest change 'is crucial for' to 'may be inevitable' to make it a little bit less sure.

Line 264 suggest 'retention' to 'retained'

Line 268 suggest change 'to shape into' to 'to aid'

[Reply and Revision]: Thank you for your positive feedback and for taking the time to review our manuscript. We appreciate your kind words about our study and are pleased to hear that you find it interesting. We are glad to learn that you are fully satisfied with the incorporation of your suggestions into the revised manuscript. Your feedback has been instrumental in enhancing the overall quality of our work. We have carefully revised these sentences following your suggestions. For detailed information, kindly refer to Lines 151, 173, 184, 201, 218, 222, 232, 267, and 271 in the revised manuscript.

Reviewer #3 (Remarks to the Author):

The authors reflected on most comments of all referees.

[Reply]: Thank you for your valuable feedback. We highly appreciate the thorough review and constructive comments from you and the other referees.

I refrain from commenting small issues, which could be still debated, modified and improved. My major concern is still the surprising lack of comments on the presumable mode of the origin of the hexaploid ancestral genome. I understand what the authors state in their rebuttal, namely that the WGT is old (65 mya) and that it is not easy to dissect the three subgenomes. Well, this is probably the case in the reshuffled genome of the mangrove species ($n=12$ chromosomes), but is this also the case in *Lagerstroemia* species with triplicates of all 8 chromosomes ($n=24$) more or less still preserved (Fig. 2e)? I still think that the authors should provide evidence for the failure of subgenome phasing (i.e. explain why this is impossible) or do this type of (standard) analysis. This is done, for example, based on gene densities in triplicated chromosome regions.

[Reply]: We appreciate the helpful comment. The hexaploid ancestral genome is generated through a phenomenon called whole-genome triplication (WGT), resulting from the hybridization between tetraploid and diploid species. This process has been documented in multiple studies¹⁷⁻¹⁹, and we concur with them. Therefore, we have mentioned it in the Introduction section with relevant citations (Line 44-45).

The whole-genome triplication (WGT) event results in the generation of three subgenomes. However, distinguishing between these subgenomes without the presence of ancestral species of tetraploid and diploid remains an unsolved challenge, especially when the WGT event is distant. The combination of different homoeologous regions in the three subgenomes remains uncertain. For instance, A₁, A₂, and A₃ represent homoeologous regions in one chromosome, and B₁, B₂, and B₃ are homoeologous regions in another chromosome. It is currently impossible to determine which A and which B will be present in the same subgenome. Meanwhile, the corresponding collinearity of WGT events has faded with time, and the distant WGT event adds another layer of complexity to subgenome phasing.

Even when simplifying the problem by utilizing homoeologous genes, we tried to perform a phylogenetic analysis to identify the outgroup of the three homoeologous copies in each species. The coding sequences were aligned using the MAFFT-PAL2NAL-Gblocks pipeline, excluding alignments shorter than 150 bp with ambiguous results and retaining the gene groups that entirely support the position between the WGT event and the speciation event. Regrettably, the majority of the 208 groups in each species lack robust bootstrap support (refer to the figure below). It indicates that this approach

is not effective either. Therefore, subgenome phasing for the mangrove tree *Sonneratia alba* or the related inland plant *Lagerstroemia speciosa* is currently unattainable. And effective subgenome phasing is feasible in species genomes that maintain relatively intact subgenomic components and undergo significant differentiation.

The distribution of bootstrap support in the phylogenetic trees for three homoeologous copies.

Two hundred bootstraps were performed, with the corresponding ortholog from *P. granatum* utilized as the outgroup. The mean values are 66.61 (left panel) and 66.46 (right panel), with corresponding median values of 65 and 64.

Nevertheless, by categorizing different-copy retention groups after the WGT event, we have illustrated the distribution of gene densities for these groups in both species (Supplementary Fig. 14). The densities of genes belonging to different-copy groups remained lower in both species, particularly in the case of three-copy groups. Additionally, we presented an expected signature of the whole-genome triplication event through collinear genes in the modern genome (Supplementary Fig. 16). This explains why the WGT event can be inferred from specific existing collinear genes, even in the presence of the rediploidization process.

References:

17. Bock, D. G., Kane, N. C., Ebert, D. P. & Rieseberg, L. H. Genome skimming reveals the origin of the Jerusalem Artichoke tuber crop species: neither from Jerusalem nor an artichoke. *New Phytologist* **201**, 1021–1030 (2014).
18. Mandáková, T., Pouch, M., Brock, J. R., Al-Shehbaz, I. A. & Lysak, M. A. Origin and evolution of diploid and allopolyploid *Camelina* genomes was accompanied by chromosome shattering. *Plant Cell* **31**, 2596–2612 (2019).
19. Aköz, G. & Nordborg, M. The *Aquilegia* genome reveals a hybrid origin of core eudicots. *Genome*

Biol **20**, 256 (2019).

Note that "gene fractionation" is mentioned in figure 3b, but I was not able to find any mention of biased or unbiased gene fractionation in the text. In fig. 3b, I did not understand what is "Fractionation, Sequence, Expression". Precisely, I did not understand to what terms "sequence" and "expression" refer.

[Reply]: We thank the reviewer for this comment. In response to this concern, we have made a modification in the revised Fig. 3 by replacing "fractionation" with "divergence" to align with the terminology used in the text.

The terms "sequence" and "expression" in Fig. 3b denote two crucial aspects of the divergence among paralogous genes. Rediploidization post polyploidization is a major process for polyploids, driving the genome toward a diploid state through divergences of homologous sequence and expression, redundancy reductions, and large chromosome rearrangements. Detail results regarding sequence divergence and expression divergence can be found in the "Divergence of WGT retained genes in the mangrove genome" section (Line 243-267).

[Revision]: The term "Fractionation" in Fig. 3b has been substituted with "Divergence".

Also, the usage of "n" and "x" is incorrect in this figure. Note that each eukaryotic organism produces haploid gametes (with n chromosomes), diploid soma has 2n chromosomes. However ploidy levels equal to x-folds (diploid 2x, triploid 3x, tetraploid 4x). Note that following the logic in the fig 2b, the outcome of diploidization is 4n (not 2n).

[Reply]: Thank you for pointing out the concern related to the usage of "n" and "x". We have made the adjustment by replacing "6n" with "6x" for the hexaploid stage in Fig. 3b.

Fig. 3b illustrates the micro perspective of the polyploidization-rediploidization process in plants. The robust support for our hypothesis is exemplified by the case of mangrove species. As the current biological definition, the mangrove species *S. alba* is a standard diploid species. This conclusion is reinforced by karyotype information (Supplementary Table 1), genome-wide Hi-C interactive heatmap (Fig. 1a), k-mer distribution (Supplementary Figure 29), and BUSCO evaluation (Supplementary Table 3). Regarding the outcome of diploidization, the rediploidization process leads to modern descendants appearing as normal diploids cytogenetically, hence the use of "2n" is more accurate than "4n".

We appreciate all the comments and suggestions.

Reviewers' Comments:

Reviewer #3:

Remarks to the Author:

I appreciate the effort made towards analyzing the WGT event in both analyzed species. Although I am not completely happy about the latest analyses and authors' response, it seems that everybody else is satisfied.

This time, I did not revise the entire ms. again, just checked the criticized figure 3B (however it should be said that it really depends on the authors whether they wish to see their work being cited and used as teaching materials - for sure, I would never use this figure in my lectures...). The authors did not get my point:

while Chr level is more or less fine because it is relatively clear what is chr rearrangement (actually rearrangementS), fusion and chr fission, it is not clear for Gene Level. Why? Gene level divergence - OK, but gene sequence? (refers to no process), gene expression? (refers to no process - there will always be some gene expression, in any organism any time). Moreover: not Nature selection, but Natural selection. Also, it is really not clear why all the ancestral chromosomes/genomes are brown, but chromosomes of the diploidized triploid genome are red and blue (this makes no sense) and it is not clear what yellow bands should show...(and nothing is explained in the legend)

Replies to reviewers' comments point by point:

Reviewer #3 (Remarks to the Author):

I appreciate the effort made towards analyzing the WGT event in both analyzed species. Although I am not completely happy about the latest analyses and authors' response, it seems that everybody else is satisfied.

[Reply]: Thank you for your valuable feedback.

This time, I did not revise the entire ms. again, just checked the criticized figure 3B (however it should be said that it really depends on the authors whether they wish to see their work being cited and used as teaching materials - for sure, I would never use this figure in my lectures...). The authors did not get my point:

while Chr level is more or less fine because it is relatively clear what is chr rearrangement (actually rearrangementS), fusion and chr fission, it is not clear for Gene Level. Why? Gene level divergence - OK, but gene sequence? (refers to no process), gene expression? (refers to no process - there will always be some gene expression, in any organism any time). Moreover: not Nature selection, but Natural selection. Also, it is really not clear why all the ancestral chromosomes/genomes are brown, but chromosomes of the diploidized triploid genome are red and blue (this makes no sense) and it is not clear what yellow bands should show...(and nothing is explained in the legend).

[Reply]: The divergence of retained genes plays a crucial role in reducing gene redundancy and serves as a primary genetic basis for genome evolution. In Fig. 3b, the terms “sequence” and “expression” represent two crucial aspects of the gene divergence. Detailed results regarding sequence divergence and expression divergence presented in the “Divergence of WGT retained genes in the mangrove genome” section suggest the potential neo- and sub-functionalization. Consequently, gene-level divergence can offer genomic evidence regarding rediploidization post polyploidization.

Meanwhile, we have made the adjustment by replacing “Nature selection” with “Natural Selection”. We also have explained the colors of chromosomes in the figure legend.

[Revision]: The brown chromosomes represent homologous chromosomes, while the red and blue chromosomes represent significantly diverged chromosomes. The yellow bands indicate regions derived from other ancestral chromosomes through chromosomal rearrangements.

We appreciate all the comments and suggestions.